# Defining basic rules for hardening influenza A virus liquid condensates

**Temitope Akhigbe Etibor[1], Silvia Vale-Costa[1], Sindhuja Sridharan[2], Daniela Brás[1], Isabelle Becher[2], Victor Hugo Mello[1], Filipe Ferreira[1], Marta Alenquer[1,3], Mikhail M Savitski[2], Maria-João Amorim[1,3]\***

[1]Cell Biology of Viral Infection Lab, Instituto Gulbenkian de Ciência, Oeiras, Portugal; [2]European Molecular Biology Laboratory, Heidelberg, Germany; [3]Cell Biology of Viral Infection Lab, Universidade Católica Portuguesa, Católica Medical School, Católica Biomedical Research Centre, Lisbon, Portugal

**Abstract** In biological systems, liquid and solid-like biomolecular condensates may contain the same molecules but their behaviour, including movement, elasticity, and viscosity, is different on account of distinct physicochemical properties. As such, it is known that phase transitions affect the function of biological condensates and that material properties can be tuned by several factors including temperature, concentration, and valency. It is, however, unclear if some factors are more efficient than others at regulating their behaviour. Viral infections are good systems to address this question as they form condensates *de novo* as part of their replication programmes. Here, we used influenza A virus (IAV) liquid cytosolic condensates, AKA viral inclusions, to provide a proof of concept that liquid condensate hardening via changes in the valency of its components is more efficient than altering their concentration or the temperature of the cell. Liquid IAV inclusions may be hardened by targeting vRNP (viral ribonucleoprotein) interactions via the known NP (nucleoprotein) oligomerising molecule, nucleozin, both *in vitro* and *in vivo* without affecting host proteome abundance nor solubility. This study is a starting point for understanding how to pharmacologically modulate the material properties of IAV inclusions and may offer opportunities for alternative antiviral strategies.

**\*For correspondence:**
mjamorim@igc.gulbenkian.pt

**Competing interest:** The authors declare that no competing interests exist.

## Editor's evaluation

Etibor and collaborators have performed a series of well-thought and careful experiments to understand some of the physical and thermodynamic properties of liquid condensates produced by the infection with the influenza A virus. However, their approach and rules could be easily applied to any other cellular phenomena that involve the formation of intracellular liquid condensates. Finally, this article is setting up the basis for an in-depth theoretical analysis of the physical phenomena described here and their correlation with the biology of intracellular liquid condensates.

## Introduction

Central to the spatiotemporal control of reactions in many viral infections is the formation of biomolecular condensates that facilitate key steps of viral lifecycles (*Etibor et al., 2021*). In influenza A virus (IAV) infection, this is key for assembling its segmented genome, a complex composed of eight different viral RNA segments (vRNA) (*Pons, 1976*). Each vRNA is encapsidated by molecules of nucleoprotein (NP) along its length, with one unit of the RNA-dependent RNA polymerase (RdRp, consisting of PB2, PB1, and PA) bound to the base-paired RNA termini, forming viral ribonucleoproteins (vRNPs) (*Alenquer et al., 2019*). How the eight vRNP complex self-assembles is unknown. It is

**eLife digest** Cells are organized into compartments that carry out specific functions. Envelope-like membranes enclose some of those compartments, while others remain unenclosed. The latter are called biomolecular condensates, and they can shift their physical states from a more liquid to a more solid form, which may affect how well they function. Temperature, molecular concentration and molecular interactions affect the physical state of condensates.

Understanding what causes physical shifts in biomolecular condensates could have important implications for human health. For example, many viruses, including influenza, HIV, rabies, measles and the virus that causes COVID-19, SARS-CoV-2, use biomolecular condensates to multiply in cells. Changing the physical state of biomolecular condensates to one that hampers viruses' ability to multiply could be an innovative approach to treating viruses.

Etibor et al. show that it is possible to harden condensates produced by influenza A virus. In the experiments, the researchers manipulated the temperature, molecular concentration and strength of connections between molecules in condensates created by influenza A-infected cells. Then, they measured their effects on the condensate's physical state. The experiments showed that using drugs that strengthen the bonds between molecules in condensates was the most effective strategy for hardening. Studies in both human cells and mice showed that using drugs to harden condensate in infected cells did not harm the cells or the animal and disabled the virus.

The experiments provide preliminary evidence that using drugs to harden biomolecular condensates may be a potential treatment strategy for influenza A. More studies are necessary to test this approach to treating influenza A or other viruses that use condensates. If they are successful, the drug could add a new tool to the antiviral treatment toolbox.

known that genomic complex formation relies on RNA-RNA interactions between distinct vRNPs and is a selective process because most virions contain exactly eight different vRNPs, as reviewed elsewhere (*Hutchinson et al., 2010*). After export from the nucleus where vRNPs are synthesised, vRNPs reach the cytosol and induce the formation of cytosolic condensates, known as viral inclusions (*Amorim et al., 2011*; *Avilov et al., 2012*; *Chou et al., 2013*; *Eisfeld et al., 2011*; *Lakdawala et al., 2014*; *Momose et al., 2011*), which we postulated to be sites dedicated to IAV genome assembly (*Alenquer et al., 2019*). Interestingly, IAV cytosolic inclusions exhibit liquid properties (fuse and divide, dissolve upon shock and are dynamic) (*Alenquer et al., 2019*), providing the first indication that defined material properties are critical for the formation of influenza epidemic and pandemic genomes.

As the list of viruses utilising liquid biomolecular condensates rapidly increases, including reoviruses, human cytomegalovirus, HIV, rabies, measles, SARS-CoV-2 (reviewed in *Etibor et al., 2021*; *Lopez et al., 2021*), it becomes pertinent to ask whether targeting the material properties could constitute a novel antiviral approach. Recently, the Sonic hedgehog pathway antagonist cyclopamine and its analogue A3E were demonstrated to inhibit human respiratory syncytial virus (hRSV) replication by altering the material properties of viral condensates (*Risso-Ballester et al., 2021*). However, compounds targeting hRSV-related (*Risso-Ballester et al., 2021*) and cancer-associated condensates exhibited off-target effects (*Klein et al., 2020*). Therefore, a critical advance in condensate disease therapy, including in the case of viral infection, requires defining how to efficiently and specifically target selected biomolecular condensates. In several studies, it was demonstrated that the properties of biomolecular condensates respond to many factors in a system-dependent manner (*Alberti et al., 2019*; *Falahati and Haji-Akbari, 2019*; *Hyman et al., 2014*; *Riback and Brangwynne, 2020*; *Mittag and Parker, 2018*; *Snead and Gladfelter, 2019*; *Milovanovic and De Camilli, 2017*; *Perdikari et al., 2020*). Entropic free energy (*Quiroz and Chilkoti, 2015*), concentration (*Riback et al., 2020*), type, number and strength of interactions (*Sanders et al., 2020*) have all been demonstrated to affect the properties of biomolecular condensates. *In vivo*, it is unknown if these changes affect equally the material properties (and function) of biomolecular condensates. This knowledge is the basis to understand which pathways could be manipulated to modulate the material properties of selected condensates, ultimately targeting their behaviour and function. For example, pathways affecting local energy production, consumption, or metabolism will alter the free energy landscape of biomolecular condensates (*Patel et al., 2017*). Similarly, pathways that

regulate the local density of condensate drivers could affect concentration (*Banani et al., 2016*; *Riback et al., 2020*). Finally, pathways regulating post-translational modifications (*Rai et al., 2018*), local pH (*Kroschwald et al., 2018*; *Munder et al., 2016*), or ionic strength (*Yang et al., 2020*), as well as strategies promoting aggregation or dissolution of condensate interactomes could affect the type, number (valency), and strength of interactions (*Bracha et al., 2019a*, *Bracha et al., 2019b*, *Zhu et al., 2019*).

Viral replication programmes rely on cellular pathways, and as such, their condensate biology is more complex than attainable with *in vitro* reconstituted systems. For this reason, studying them in their native intracellular environment will more accurately define how they respond to specific stimuli. Our study depends on introducing a disturbance and evaluating its effect on the material properties through two different approaches. The first is to map the intracellular biophysical traits of the perturbations on viral condensates measuring thermodynamic parameters, as done in *Riback et al., 2020*; *Shimobayashi et al., 2021*; *Wei et al., 2020*. As liquids tend to be spherical to minimise surface tension (*Elbaum-Garfinkle et al., 2015*; *Lee et al., 2023*), we assessed their morphology and topology, and as molecules move freely in liquids, we also measured the Gibbs free energy of partition (henceforth called free energy, ΔG) to define the degree of molecular stabilisation (*Shin et al., 2017*; *Riback et al., 2020*). The lower the value of ΔG, the more stabilised the system becomes. The calculation of ΔG requires assessing the ratio of concentration of material inside a condensate ($C_{dense}$), relative to the concentration dissolved in the cytosol ($C_{dilute}$), which provides the value of the partition coefficient (K), and allows the use of the formula $\Delta G = -RT \ln K$ and evaluate how the system progresses. In addition, the nucleation density determines how many viral condensates are formed per area of cytosol. Overall, the data will inform us if changing one parameter, e.g. the concentration, drives the system towards condensates of different sizes with the same or more stable properties (e.g. induced by phase transitions), or changes their abundance on account of additional available nucleation centres or dissolves the condensates (*Riback et al., 2020*; *Snead et al., 2022*). These are the most relevant parameters for our scientific question, which is to define how IAV liquid viral inclusions can be hardened. However, other values are retrieved from these analyses, revealing, for example, the types of interactions required to maintain the properties of the system. This can be illustrated by reports showing that with bulk concentration, $C_{dilute}$ is constant in a binary mixture (*Klosin et al., 2020*) but increases in multi-component systems (*Riback et al., 2020*). This type of information has implications about the condensates formed during influenza infection. If the system is binary, the eight vRNPs could behave as a single component, on account of their similar structure, and interact with the host Ras-related protein in the brain 11 (Rab11), reported to be part of inclusions (*Avilov et al., 2012*; *Eisfeld et al., 2011*; *Amorim et al., 2011*). If IAV inclusions are multi-component systems, each vRNP could behave as an independent entity, on account of differences in length, RNA sequence, and valency or, as an alternative, viral inclusions could contain yet unidentified components other than Rab11 and vRNPs. The second type of approach to accurately define how IAV liquid inclusions respond to specific stimuli is to assess the effect of the perturbations in the dynamics and kinetics of viral condensates, thus inferring changes in elasticity and viscosity (*Wang et al., 2022*; *Alberti et al., 2019*). This relies on using live-cell imaging approaches to assess if two condensates can fuse, internally rearrange, or exchange material, which are all *bona fide* traits of liquids but not of solids (*Wang et al., 2022*; *Alberti et al., 2019*). The information collected will offer insight into how the material properties of IAV inclusions are regulated and maintained. In addition, understanding how IAV viral inclusions can be hardened can be therapeutically relevant if they rely on their liquid character for function.

The objective of this project is to determine the most effective and precise approaches for hardening IAV liquid inclusions. We find that the stabilisation of intersegment interactions is more efficient at hardening IAV inclusions than varying the temperature or the concentration of the drivers of IAV inclusions. Importantly, we show that the hardening topological phenotype is observed in the lungs of infected mice. We also report that it is possible to affect viral inclusions without imposing additional changes in host protein abundance and solubility using solubility proteome profiling (SPP) of infected cells (*Sridharan et al., 2022*). In sum, our data support the development of strategies targeting the material properties of cellular condensates in viral infections and provide a critical advance in how these structures may be regulated.

## Results

### Framework to identify perturbations that harden IAV liquid inclusions

We previously demonstrated that viral inclusions formed by IAV infection display a liquid profile in the sense that they drip, acquire a spherical shape upon fusion, and dissolve in response to hypotonic shock or brefeldin A treatment (*Alenquer et al., 2019*). Here, we seek to identify the best strategies to harden viral inclusions to investigate if altering their material properties may be a novel antiviral therapy. For this, we systematically probed and compared the impact of temperature, concentration, and number/strength of ligations on the material properties of liquid viral inclusions, as a proxy of entropic, molecular, and valency contributions, respectively. We selected these parameters given the deep understanding on how they regulate the interactions amongst components and the material properties of condensates (*Riback et al., 2020*; *Sanders et al., 2020*; *Quiroz and Chilkoti, 2015*; *Figure 1A*). Methodologically, we employed established protocols for imposing the selected thermo-dynamic perturbations. We quantified the impact of these perturbations on the number, nucleation density ($\rho = \frac{number\ of\ inclusion}{Cytoplasm\ Area}$, µm$^{-2}$), size, shape, dynamics, supersaturation ($S = In\frac{Cdilute}{Csaturation}$, in which $C_{saturation}$ is the concentration above which molecules demix from an homogenous system), and the Gibbs free energy of partition (ΔG = -RT$In$K, in which $K = \frac{Cdense}{Cdilute}$ is the partition coefficient) to define how our system adapts to the perturbations. Material concentrations inside ($C_{dense}$) and outside ($C_{dilute}$) viral inclusions were measured using the analytical strategies described in *Riback et al., 2020*; *Shimobayashi et al., 2021*, and shown in *Figure 1B* (and validated as described in Materials and methods and in *Figure 2—figure supplement 1*). For this, we used the mean fluorescence intensity (MFI) of NP as proxy of vRNP concentration (*Amorim et al., 2011*; *Vale-Costa et al., 2016*), as it is well established that the majority of cytosolic NP is in the form of vRNPs (*Avilov et al., 2012*; *Momose et al., 2011*; *Eisfeld et al., 2011*; *Amorim et al., 2011*).

Our goal was to identify which perturbations translated into significant shifts in ΔG to further explore whether these resulted in dramatic alterations in the material properties of viral inclusions, by assessing their kinetics and dynamics (*Figure 1C*) and determine how they impact viral replication *in vivo* (*Figure 1D*).

### Changes in temperature mildly perturb IAV inclusions

Cellular steady state is maintained at a narrow permissive physiological range, including of temperature. However, biomolecular condensates respond to fluctuations in temperature, and we took advantage of this to assess the entropic contribution of free energy and evaluate whether regulating host cell metabolism could offer future solutions to harden IAV liquid inclusions (*Figure 2A*). We quantitatively analysed the viral inclusions formed in cells incubated at 4°C, 37°C, and 42°C for 30 min at 8 hr post-infection (hpi) (representative images in *Figure 2B*). This short duration in temperature shift did not alter the levels of cytosolic vRNPs, as expected (*Figure 2C*). Increasing the temperature from 37°C to 42°C did not significantly change the size (*Figure 2D*), aspect ratio, or number of viral inclusions (*Figure 2D–G*), but decreased the concentration of vRNPs in condensates ($C_{dense}$), and increased the nucleation density despite not altering the concentration of vRNPs in the milieu ($C_{dilute}$) (*Figure 2H–M*, *Supplementary file 1* (Sheet 1)). This means that increasing the temperature up to 42°C still maintains the system in a two-phase regime, but affects the nucleation capacity, increasing the number of inclusions. Of note, vRNPs become homogenously distributed in the cytosol at 43.5°C (data not shown), pointing that the entire system may undergo regulatory processes. Importantly, this increase in temperature modestly destabilised the structure, as observed by an increase in Gibbs free energy (–2167.3±2361 J/mol @ 37°C to –1477.9±228 J/mol @ 42°C, mean ± SD, *Figure 2N–O*, *Supplementary file 1* (Sheet 1)). Conversely, decreasing the temperature until 4°C leads to an increase in the size of inclusions that is statistically significant considering 42°C to 4°C shifts only (shift in area from 0.2896±0.02 µm$^2$ at 42°C to 0.3474±0.05 at 4°C), rounds up liquid inclusions, and decreases their nucleation capacity and abundance (the latter significant only considering 42°C and 4°C, *Figure 2D–H* and *Supplementary file 1* (Sheet 1)). A drop in temperature increases the concentration of vRNPs in inclusions ($C_{dense}$ at 37°C of 3116.0±0.05 AU, mean ± SD, and at 4°C of 2144.5±0.04 AU, *Supplementary file 1* (Sheet 1)), and does not significantly change the stability of IAV inclusions as determined by Gibbs free energy (–2415.6±273 J/mol @ 4°C, *Figure 2N and O*). Overall, the data indicate that the temperature increasing from 4°C to 42°C shifts our system to smaller inclusions that have less vRNPs. $C_{dilute}$ did not change but there is an increase in nucleation density which indicates that heat disruption

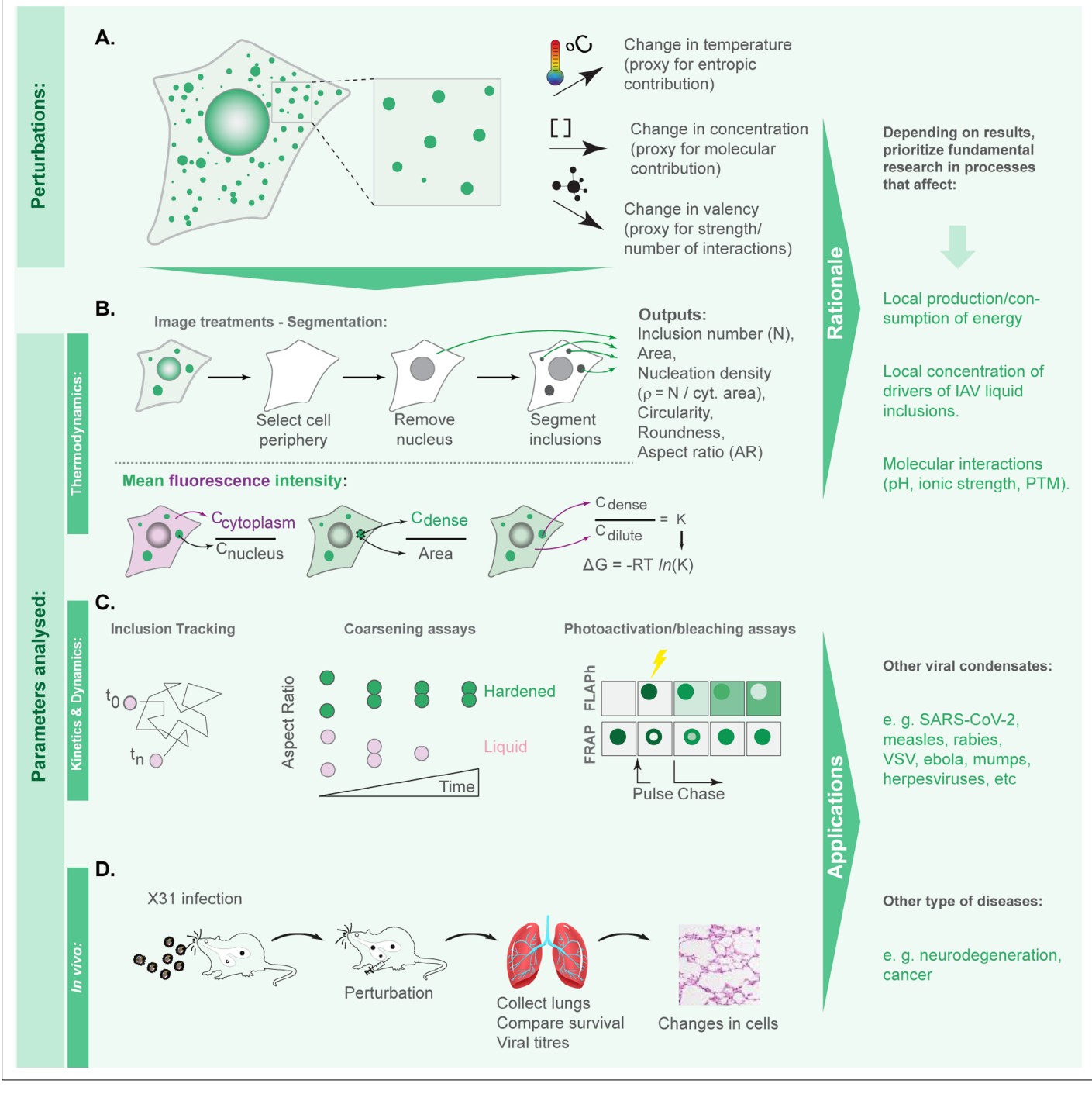

**Figure 1.** Framework applied in this study to define the hierarchy in effectiveness in hardening influenza A virus (IAV) liquid inclusions or other condensates. (**A**) To compare the contributions of entropy, concentration, and valency/strength/type of interactions, we subjected infected cells to the different perturbations: temperature, concentration of viral inclusion drivers (viral ribonucleoproteins [vRNPs] and Ras-related in brain 11a [Rab11a]) and number or strength of interactions between different vRNPs using the well-studied vRNP pharmacological modulator, nucleozin, that increases the number and strength of intersegment interactions. (**B**) Our aim is to determine which amongst these perturbations impact more dramatically the material properties of viral inclusions and for this we measured number, shape, size, and Gibbs free energy of partition (free energy, ΔG). Methodologically, we segmented circa 20 cells under the different conditions to measure the above-mentioned parameters and the amount of material inside ($C_{dense}$) and outside ($C_{dilute}$) viral condensates. With this, we calculated the partition coefficient K and extrapolated the ΔG. (**C**) When ΔG dramatically changed, we assessed how perturbations altered the material properties of IAV inclusions by comparing how fast and how much they moved (using coarsening assays, particle tracking, photobleaching to inspect internal rearrangements, and fluorescence loss after photoactivation [FLAPh]). (**D**) We also assessed whether the phenotype could be recapitulated *in vivo* using mice infected with influenza A virus reassortant X31. The

*Figure 1 continued on next page*

overall goal of this framework is to determine, for IAV, how liquid inclusions may be efficiently hardened to prioritise research and development of strategies with that activity. Additionally, the framework may be applied to other systems, including other viruses, for informed decisions on how to harden condensates.

---

of weak molecular interactions leads to alterations in nucleation, fusion, and fission, as reported previously (*Iserman et al., 2020*). However, we did not observe significant alterations in the stabilisation of our system, supporting that alterations in the conditions tested for temperature do not affect the material properties of viral inclusions.

## Changes in concentration of viral inclusions' drivers do not impact their liquid profile

Two factors were shown to drive the formation of IAV inclusions – vRNPs and Rab11a (*Amorim et al., 2011*; *Eisfeld et al., 2011*; *Lakdawala et al., 2014*; *Vale-Costa et al., 2016*; *Alenquer et al., 2019*; *Veler et al., 2022*). In fact, vRNP accumulation in liquid viral inclusions requires its association with Rab11a directly via the viral polymerase PB2 (*Veler et al., 2022*; *Amorim et al., 2011*), and the liquid character may be maintained by an incompletely understood network of intersegment interactions bridging several cognate vRNP-Rab11 units on flexible membranes (*Vale-Costa et al., 2016*). As the concentration of material is a key determinant for the physical properties of condensates (*Riback et al., 2020*; *Weber and Brangwynne, 2015*; *Hernández-Vega et al., 2017*), we evaluated how concentration of these two drivers impacts the behaviour of IAV inclusions.

For this, we took advantage of the fact that vRNP levels increase during infection (*Kawakami et al., 2011*), and analysed viral inclusions over a time course, in two conditions: with endogenous levels of Rab11a (using cells expressing GFP, as in *Alenquer et al., 2019*), and overexpressing Rab11a (in the form of GFP-Rab11a, as in *Alenquer et al., 2019*; *Figure 3A–B*, *Figure 3—figure supplement 2*). With this approach, we aimed at analysing whether the material properties of viral inclusions changed over time and whether increasing the levels of Rab11 would alter these properties. This strategy would reveal if regulating Rab11a activity could harden IAV liquid inclusions.

In GFP expressing cells, as the progeny vRNP pool reaches the cytosol (*Figure 3B and C*), viral inclusions augment in size (from $0.172\pm0.04$ to $0.289\pm0.06$ $\mu m^2$, mean $\pm$ SD, *Figure 3D*), with similar aspect ratio (*Figure 3—figure supplement 2A and B*). There is a mild reduction in the number of inclusions from 8 hpi onwards, as measured by the nucleation density ($\rho$) (*Figure 3E*, *Figure 3—figure supplement 2C*, all topological data in *Supplementary file 1* (Sheet 2)). As infection progresses, the concentration of vRNPs inside condensates increases until 8 hpi (*Figure 3F* and *Figure 3—figure supplement 2D and E*), accompanied by an increase in the diluted cytosolic phase (*Figure 3G* and *Figure 3—figure supplement 2D and F*, *Supplementary file 1* (Sheet 2)), and both parameters stabilise thereafter, indicating that the critical concentration occurs around 8 hpi. This indicates that the liquid inclusions behave as a multi-component system and allow us to speculate that the differences in length, RNA sequence, and valency that each vRNP may be key for the integrity and behaviour of condensates. Importantly, Gibbs free energy (normalised to 3 hpi) is lowest at 6 hpi ($-1799.0\pm623$ J/mol) and destabilises mildly onwards ($-1139.8\pm382$, $-1131.2\pm444$, and $-833.8\pm342$ J/mol @ 8, 12, and 16 hpi, respectively) (*Figure 3H*, *Figure 3—figure supplement 2G, H*, *Supplementary file 1* (Sheet 2)). These results are consistent with the increase in cytosolic vRNP leading to bigger sized inclusions that overall maintain the same concentration although becoming modestly destabilised, suggesting that the material properties are also modestly affected. When overexpressing Rab11a (right side of each graph), cytosolic vRNPs also accumulated in viral inclusions that increased with infection (*Figure 3C–D*, from $0.243\pm0.03$ to $0.385\pm0.04$ $\mu m^2$), but were significantly bigger than viral inclusions in GFP expressing cells. In addition, the nucleation density was higher (*Figure 3E* and *Figure 3—figure supplement 2C*), despite having similar aspect ratio (*Figure 3—figure supplement 2A and B*), $C_{dense}$ (*Figure 3F* and *Figure 3—figure supplement 2D and E*) and $C_{dilute}$ (*Figure 3G*, *Figure 3—figure supplement 2D and F*). The lowest value of Gibbs free energy occurs at 8 hpi ($-1337.7\pm331$ J/mol) and destabilises from then onwards ($-1145.3\pm443$ and $-895.3\pm394$ J/mol @ 12 and 16 hpi, respectively, *Figure 3H*, *Figure 3—figure supplement 2G and H*, all thermodynamic data in *Supplementary file 1* (Sheet 3)). This is consistent with Rab11a overexpression giving rise to bigger viral inclusions than cells expressing native levels of Rab11a that overall contained the same vRNP

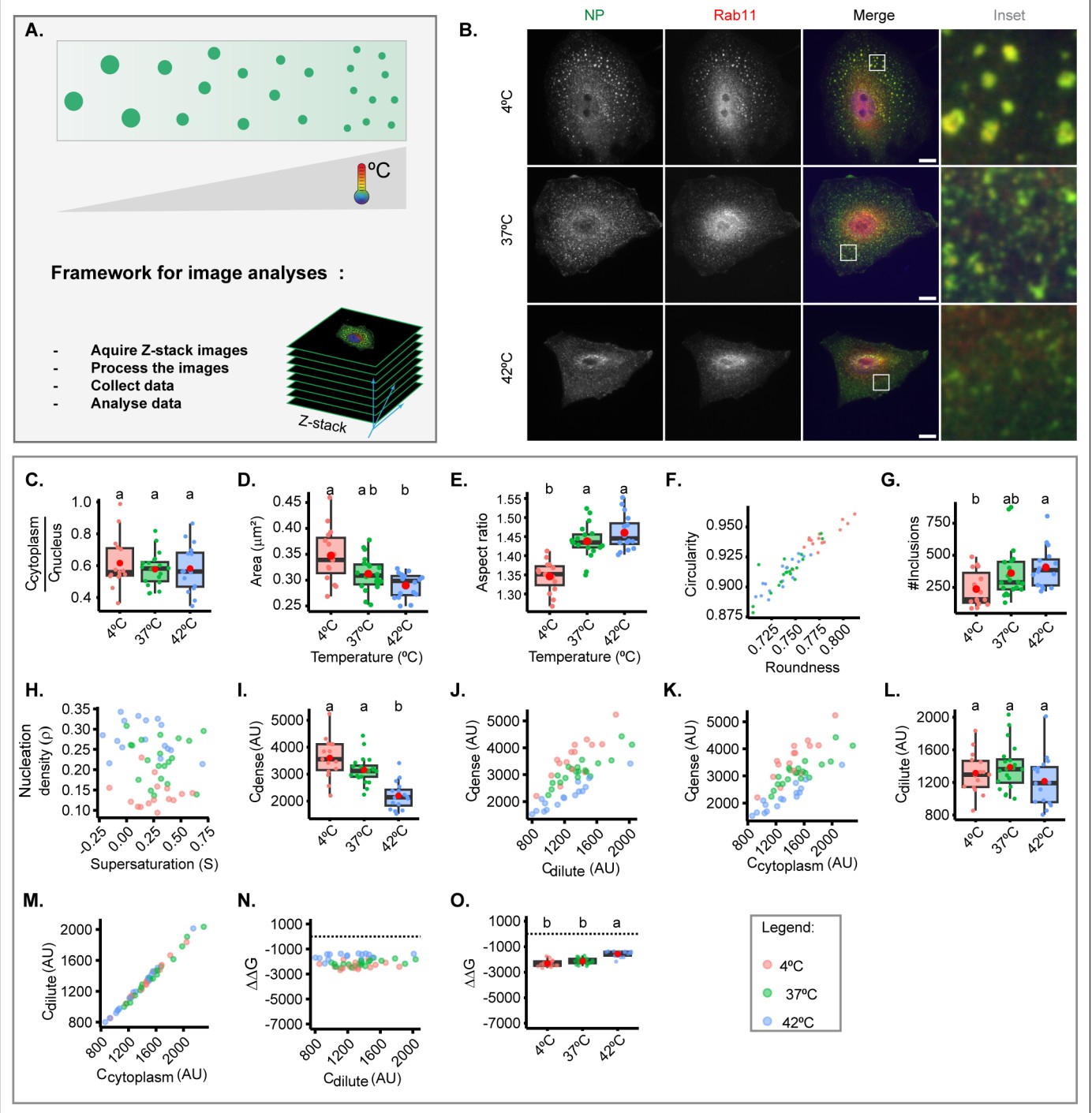

**Figure 2.** Thermal changes mildly perturb the material properties of inclusions. A549 were infected at a multiplicity of infection (MOI) of 3 with PR8 virus for 8 hr, incubated at different temperatures (4°C, 37°C, 42°C) for 30 min, fixed, and analysed by immunofluorescence using antibody staining against Rab11 and nucleoprotein (NP) as a proxy for viral ribonucleoprotein (vRNP). The biophysical parameters were extracted from immunofluorescence images (n=15–20), adapting the method published by *Riback et al., 2020*; *Shimobayashi et al., 2021*, to determine concentration $C_{dense}$ as the mean fluorescence intensity of vRNPs in the segmented influenza A virus (IAV) inclusions, while concentration $C_{dilute}$ was extrapolated from the cytoplasmic vRNP intensity outside the inclusions. Each dot is the average value of a measured parameter within or outside IAV inclusions per cell. Also, size and shape of inclusion were extracted from inclusions after image segmentation. Parameters that were normalised to an infection state without IAV inclusions (3 hr post-infection [hpi]) are indicated by a dashed horizontal line. Above each boxplot, same letters indicate no significant difference between them, while different letters indicate a statistical significance at α=0.05. All data are displayed in *Supplementary file 1* (Sheet 1). Abbreviations: AU, arbitrary unit. (**A**) Representative depiction of the experimental analysis workflow. (**B**) Representative images of fixed A549 cells

*Figure 2 continued on next page*

*Figure 2 continued*

infected with PR8 virus showing alterations in viral inclusions at different temperatures. (**C**). Boxplot depicting the fold change in cytoplasmic to nuclear vRNP concentration; p=0.684 by one-way ANOVA followed by Tukey multiple comparisons of means. (**D**) Boxplot of mean viral inclusion area ($\mu m^2$) per cell; p=0.00234 by Kruskal Wallis Bonferroni treatment. (**E**) Boxplot of aspect ratio of inclusion; p<0.001 by one-way ANOVA followed by Tukey multiple comparisons of means. (**F**) Scatter plot of inclusions circularity versus roundness. (**G**) Boxplot showing number of viral inclusions per cell; p<0.001 by one-way ANOVA, followed by Tukey multiple comparisons of means. (**H**) Scatter plot of nucleation density ( $\rho = \frac{number\ of\ inclusion}{cytoplasm\ Area}$ $\mu m^{-2}$) versus degree of supersaturation (S=In $\frac{Cdilute}{Csat}$), as a measure of propensity to remain dispersed in the cytoplasm. (**I**) Boxplot of of vRNP concentration within inclusions ($C_{dense}$ (AU)) p<0.001 by one-way ANOVA, followed by Tukey multiple comparisons of means. (**J**) Scatter plot of $C_{dense}$ (AU) versus surrounding cytoplasm ($C_{dilute}$, AU). (**K**) Scatter plot $C_{dense}$ (AU) versus its total cytoplasmic vRNP concentration ($C_{cytoplasm}$, AU). (**L**) Boxplot showing $C_{dilute}$ (AU); p=0.203 by one-way ANOVA followed by Tukey multiple comparisons of means. (**M**) Scatter plot of $C_{dilute}$ (AU) versus $C_{cytoplasm}$ (AU). (**N**) Scatter plot of fold change in free energy of partition ($\Delta\Delta G$, J/mol) where $\Delta G$ = -RTInK, and K = ($\frac{Cdense}{Cdilute}$), and $\Delta\Delta G = \Delta G - \Delta G_{3\ hpi}$, versus $C_{dilute}$ (AU). (**O**) Boxplot of $\Delta\Delta G$ (J/mol); p<0.001 by one-way ANOVA followed by Tukey multiple comparisons of means.

The online version of this article includes the following figure supplement(s) for figure 2:

**Figure supplement 1.** Validation of method analysing thermodynamics parameters.

concentration and destabilise slightly later in the time course of infection, confirming that Rab11a is involved in nucleating and/or maturating viral inclusions. Therefore, these data indicate that altering the concentration of vRNPs and/or Rab11a affects the size but modestly impact IAV inclusions' material properties.

## The increase in type/strength of vRNP interactions dramatically stabilises IAV inclusions

Another critical regulator of condensate properties is the type and strength of interactions among its components (*Alberti and Hyman, 2021*). Therefore, we predict that oligomerising vRNPs to each other, or to Rab11a, will change the viscoelasticity of condensates in similar manner to iPOLYMER in intracellular hydrogels (*Nakamura et al., 2018*). For IAV, it was shown by many independent groups that the drug nucleozin operates as a pharmacological modulator that oligomerises (sticks together) all forms of NP (*Amorim et al., 2013*; *Nakano et al., 2021*; *Kao et al., 2010*). In fact, it was demonstrated that this drug has affinity for three different sites in NP (*Kao et al., 2010*) chemically polymerising NP either free or in vRNPs in a reversible manner (*Amorim et al., 2013*). Interestingly, nucleozin was described as a novel class of influenza antivirals targeting the viral protein NP, potently inhibiting IAV replication in cultured cells and in a mouse model of influenza infection (*Cianci et al., 2012*). However, it readily evolved escape mutant viruses carrying the single substitution tyrosine to histidine in position 289 of NP (NP-Y289H) (*Kao et al., 2010*). Despite its capacity to evolve resistance, our strategy is to take advantage of a well-known tool to probe the effects of increasing the number and type of intra- and inter-vRNP interactions in the material properties of IAV inclusions (*Figure 4A*).

With this reasoning, we evaluated the thermal stability of inclusions in the presence or absence of nucleozin in order to confirm its pharmacological modulator activity on liquid viral inclusions (*Sridharan et al., 2019*). It is well established that increasing temperature shifts a thermodynamic system to a homogeneous mix. In agreement, when we exposed IAV infected cells to a range of temperatures (4°C, 37°C, and 42°C), we found that higher temperatures yield smaller inclusions (*Figure 2*, *Figure 4—figure supplement 2*) during the two-phase regime, but tending towards its homogenous distribution in the cytoplasm that was achieved at 43.5°C. Interestingly, when infected cells were exposed to the same thermal conditions after nucleozin treatment, inclusions were irresponsive to thermal fluctuation, which indicates that nucleozin stabilises viral inclusions (*Figure 4—figure supplement 2*).

Next, we tracked how nucleozin affected IAV liquid inclusions, by imposing the infected cells to this drug for different periods ranging from from 5 min to 2 hr. We observed that nucleozin-treated inclusions form a multi-shaped meshwork unlike the rounded liquid droplets formed without nucleozin (*Figure 4B*) and confirmed that nucleozin does not affect Rab11a localisation in uninfected cells (*Figure 4P*). Nucleozin affected the concentration of vRNPs in the cytosol that decreased with the time of treatment (*Figure 4C*), presumably by blocking vRNP nuclear export and/or changes accessibility of antibodies to oligomerised NP. Conversely, nucleozin treatment increased the size of viral inclusions (from 0.284±0.04 without nucleozin to 1.02±0.18 $\mu m^2$ with 2 hr treatment, *Figure 4D*), which lost circularity (0.893±0.02 without nucleozin to 0.761±0.02 2 hr treatment) and roundness (0.734±0.01

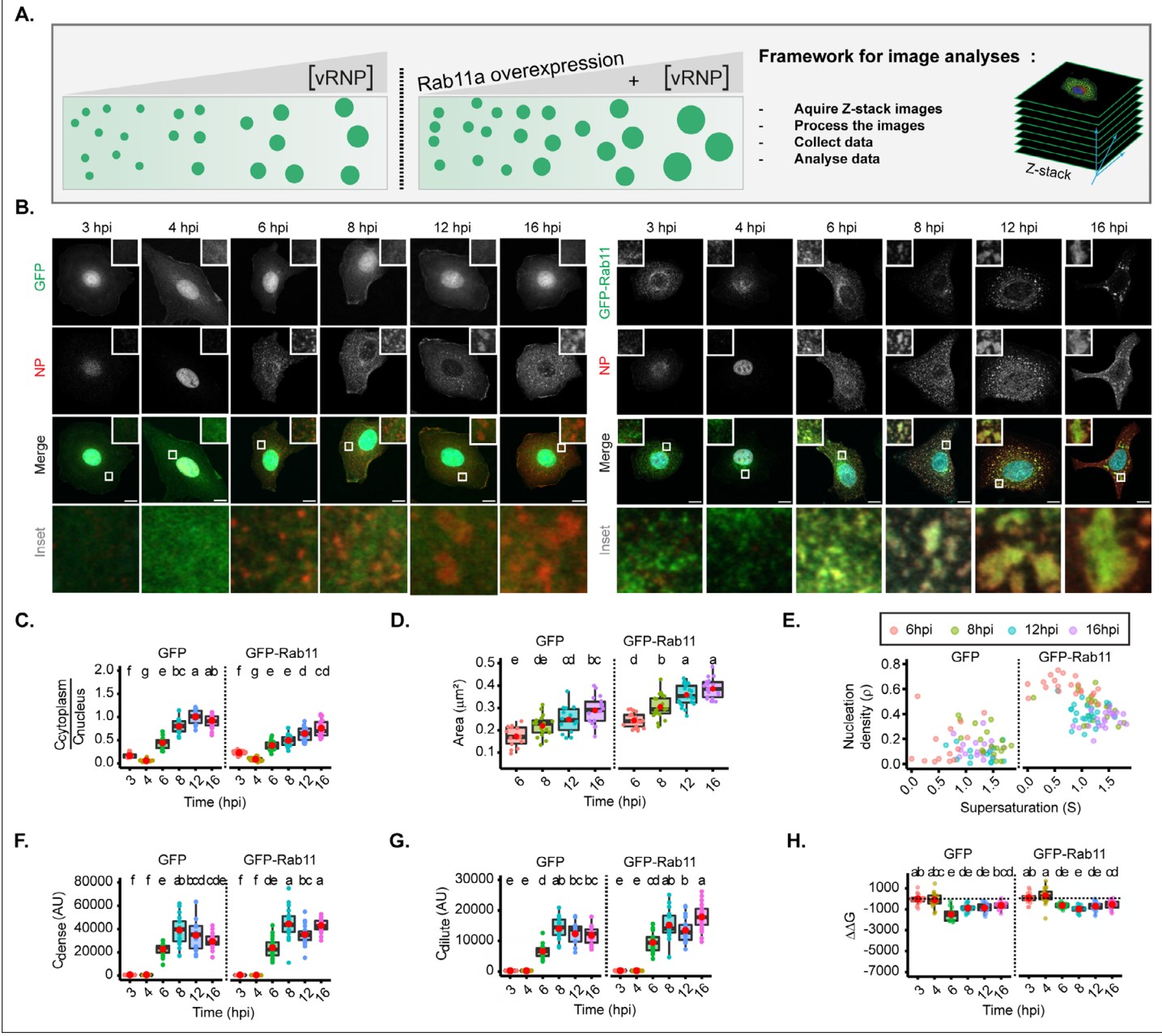

**Figure 3.** Changes in concentration of viral ribonucleoproteins (vRNPs) and Ras-related in brain 11a (Rab11a) modestly alter the material properties of viral inclusions. (A–H) A549 cells stably expressing GFP, or GFP-Rab11a-WT were infected at an MOI of 3 with PR8 virus and, at the indicated timepoints, were fixed, and analysed by immunofluorescence using an antibody against nucleoprotein (NP) (as a proxy for vRNPs). (Number of cells (n)=14–18 for GFP and 19–25 for GFP-Rab11-WT). (C–H) Each dot is the average value of measured parameters per cell. Above each boxplot, same letters indicate no significant difference between them, while different letters indicate a statistical significance at α=0.05 using one-way ANOVA, followed by Tukey multiple comparisons of means for parametric analysis, or Kruskal Wallis Bonferroni treatment for non-parametric analysis. All thermodynamic related values are displayed in ***Supplementary file 1*** (Sheets 2 and 3). Abbreviations: AU, arbitrary unit. (**A**) Representative depiction of the experimental analysis workflow. (**B**) Immunofluorescence images of infected cells at different hours post-infection (hpi) in cells overexpressing GFP (left) or GFP-Rab11 (right) (both in green); NP (red, as a proxy of vRNPs), and nucleus (blue). Scale bar = 10 μm. (**C**) Boxplot depicting the fold change in the ratio of cytoplasmic to nuclear vRNPs concentration at different times of infection, with endogenous or overexpressed Rab11a; p<0.001; Kruskal Wallis Bonferroni treatment. (**D**) Boxplot of mean inclusion area (μm²) per cell; p<0.001 by one-way ANOVA, followed by Tukey multiple comparisons of means. (**E**) Scatter plot showing nucleation density ($\rho$, μm⁻²) versus degree of supersaturation (S). (**F**) Boxplot of $C_{dense}$ (AU); p<0.001 by Kruskal Wallis Bonferroni treatment. (**G**) Boxplot of $C_{dilute}$ (AU); p<0.001 by Kruskal Wallis Bonferroni treatment. (**H**) Boxplot of ΔΔG (J/mol); p<0.001 by Kruskal Wallis Bonferroni treatment. Conditions were normalised to an infection state without IAV inclusions (3 hpi) that is indicated by the dashed black line.

The online version of this article includes the following figure supplement(s) for figure 3:

*Figure 3 continued on next page*

*Figure 3 continued*

**Figure supplement 1.** Validation of method analysing thermodynamics parameters.

**Figure supplement 2.** Change in viral ribonucleoprotein (vRNP) and Ras-related in brain 11a (Rab11a) concentration modestly alter inclusions properties.

without nucleozin to 0.672±0.02 with 2 hr treatment, *Figure 4E–F*) and decreased in number (from 310.5±133 to 38.1±34 after 2 hr treatment, *Figure 4G and H*), suggesting that they were stiffer. Interestingly, $C_{dense}$ increased dramatically (from 2125.8±0.09 without nucleozin to 3650.0±0.03 with 2 hr nucleozin), *Figure 4I–K*) and $C_{dilute}$ decreased and became stable after 20 min treatment (from 728.1±213 without nucleozin to 398.6±94 after 2 hr treatment, *Supplementary file 1* (Sheet 4, total $C_{dilute}$), *Figure 4J and L–M*), suggesting a decrease in $C_{sat}$ (saturation concentration), that is compatible with the formation of a stable complex rather than a liquid condensate. Importantly, these structures were energetically more stable, with lower free energy (from –1711.1±397 J/mol without nucleozin to –5388.4±808 J/mol 2 hr post-nucleozin addition (*Figure 4N–O*, all topological and thermodynamic values in *Supplementary file 1* (Sheet 4)).

Using the knowledge that the viral protein NP is critical to vRNP-nucleozin binding, and that the NP-Y289H renders the virus resistant, we sought to validate our findings with the mutant after adding nucleozin for 1 hr at 8 hpi. Immunofluorescence microscopy confirmed that nucleozin did not affect Rab11a in uninfected cells (mock), aggregated NP, and Rab11a in cells infected with the WT PR8 virus and that the cytoplasmic vRNP aggregation did not occur in cells infected with the NP-Y289H virus mutant (*Figure 4P*). Furthermore, these observations were validated by calculating the area of viral inclusions (from 0.16 to 0.18 μm² in all mock conditions, to 0.26–0.28 μm² upon infection both with WT and NP-Y289H mutant virus). With nucleozin, the size of viral inclusions with WT virus increased to 0.81±0.22 μm² and this increase was not observed with the mutant and maintained 0.27±0.04 μm² (*Figure 4Q*). The same was observed for nucleozin stabilisation, as measured by Gibbs free energy that was only seen with the WT PR8 virus and not with NP-Y289H, demonstrating that viral inclusion stiffness related to the viral protein NP and its effect on vRNP-vRNP interactions (*Figure 4R*).

Together, the data suggest that stabilising vRNP interactions changes inclusions more efficiently than the other strategies tested above.

## Modifiers of strength/type of interactions among vRNPs but not concentration or temperature harden liquid IAV inclusions

Changing the strength of interactions amongst vRNPs impacted viral inclusions' thermodynamics the most and our assumption is that this translated into modification in the material properties.

To validate our assumption, we compared the material properties of viral inclusions arising in infected cells with and without overexpression of Rab11a, with those subjected to changes in strength/type of interactions using nucleozin. Our prediction is that in the first case, we preserved the liquid character whilst in the second, inclusions were stiffer. In addition, it would be possible to accurately determine if overexpressing Rab11a impacts in material properties.

We first assessed the material properties of viral inclusions upon changes in concentration of Rab11a. Initially, we tested whether, in the two conditions, viral inclusions could fuse and divide. We observed that both fusion and fission events were taking place at late timepoints (*Figure 5A*, *Figure 5—video 1*, *Figure 5—video 2*, *Figure 5—video 3*, *Figure 5—video 4*). In a second approach, we measured the internal rearrangement in viral inclusions. Because of the small size and highly dynamic nature of IAV inclusions, previous attempts to perform fluorescence recovery after photobleaching experiments resulted in highly variable recovery rates (*Alenquer et al., 2019*; *Amorim et al., 2011*) that were unable to accurately determine if internal rearrangements were taking place viral inclusions. As the microtubule depolymerising drug nocodazole largely blocks the movement of IAV inclusions, rendering them larger and more spherical (*Amorim et al., 2011*; *Avilov et al., 2012*), we opted for bleaching IAV inclusions upon treating them with nocodazole (*Figure 5B*, *Figure 5—videos 5; 6*). We observed that at late stages of infection, vRNPs were able to internally rearrange in normal conditions as well as when Rab11a was overexpressed (*Figure 5C*). These two experiments were important to validate that concentration did not impact the material properties of viral inclusions. However, to establish if Rab11a overexpression had any impact in the dynamics of viral inclusions, we tested the

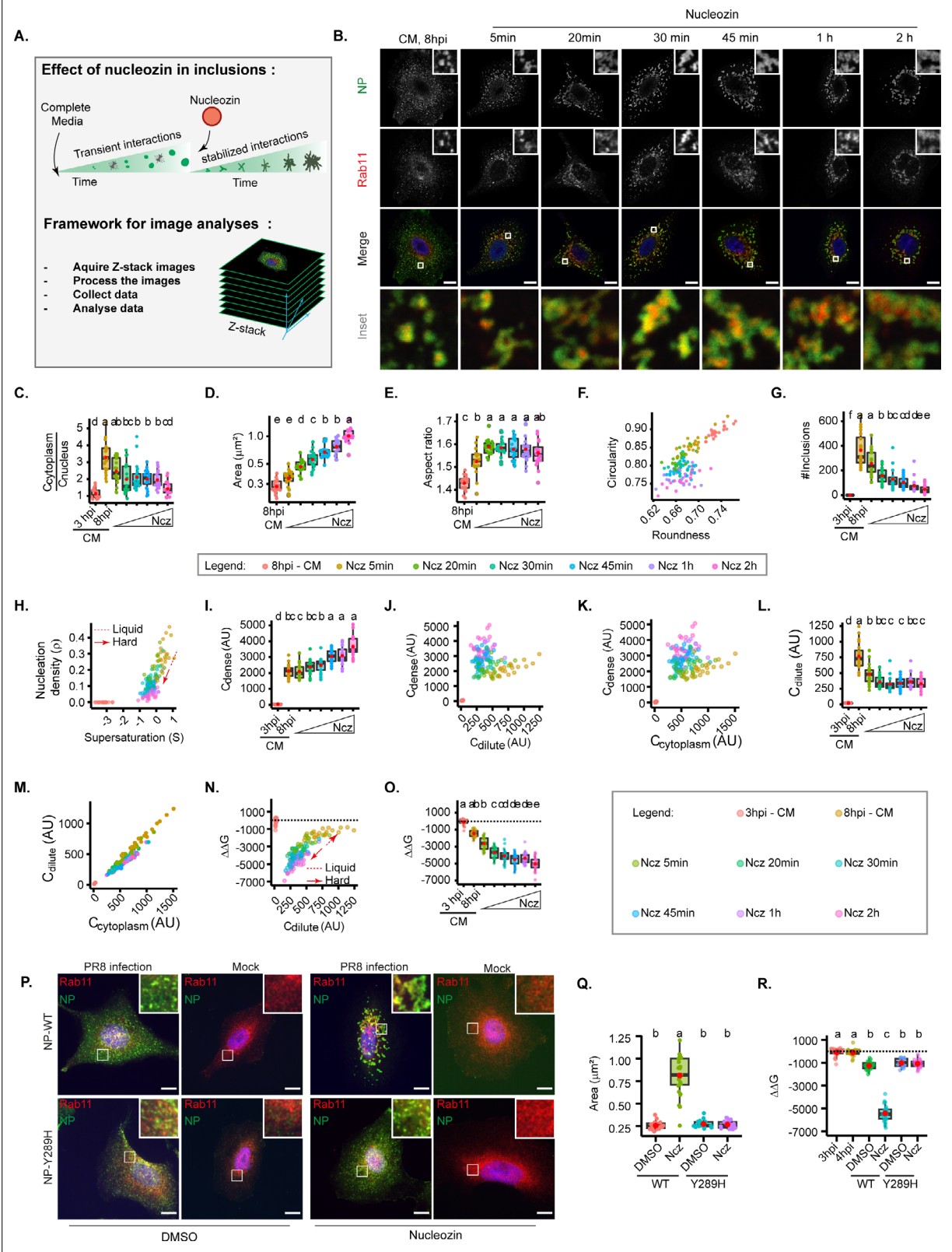

**Figure 4.** Increasing interaction number and strength stabilises influenza A virus (IAV) inclusions. A549 cells were infected at an MOI of 3 with PR8 virus for 8 hr, then incubated with 5 µM of nucleozin (Ncz), a viral ribonucleoprotein (vRNP) pharmacological modulator, for different time periods from 5 min to 2 hr, before fixing. Cells were processed for immunofluorescence analysis, using antibodies against nucleoprotein (NP) and Ras-related in brain 11a (Rab11a) (n=20–21). Each dot is the average value of a measured parameter per cell. Conditions normalised to an infection state without IAV inclusions

*Figure 4 continued on next page*

*Figure 4 continued*

(3 hr post-infection [hpi]) are indicated by a dashed black horizontal line. Above each boxplot, same letters indicate no significant difference between them, while different letters indicate a statistical significance at α=0.05 using one-way ANOVA, followed by Tukey multiple comparisons of means for parametric analysis, or Kruskal Wallis Bonferroni treatment for non-parametric analysis. All the values calculated for the thermodynamics parameters have been included as *Supplementary file 1* (Sheet 4). Abbreviations: AU, arbitrary unit, CM, complete media, and Ncz, nucleozin. (**A**) Representative depiction of the experimental and analysis workflow. (**B**) Representative images of infected A549 cells subjected (or not) to increasing periods of Ncz treatment. NP (green), Rab11a (red), and nucleus (blue). Scale bar = 10 μm. (**C**) Boxplot depicting the fold change in the ratio of cytoplasmic to nuclear vRNPs concentration before and after Ncz treatment at 8 hpi; p=6.16e-14 by Kruskal Wallis Bonferroni treatment. (**D**) Boxplot of mean inclusion area per cell; p<0.001 by Kruskal Wallis Bonferroni treatment. (**E**) Boxplot of inclusion aspect ratio; p<2e-16 by Kruskal Wallis Bonferroni treatment. (**F**) Scatter plot of inclusion circularity versus roundness. (**G**) Boxplot showing the number of inclusions per cell; p<0.001 by Kruskal Wallis Bonferroni treatment. (**H**) Scatter plot of nucleation density ( $\rho$ , μm$^{-2}$) versus degree of supersaturation (S). (**I**) Boxplot showing increasing inclusion $C_{dense}$ (AU) with increasing Ncz incubation period; p<0.001 by Kruskal Wallis Bonferroni treatment. (**J**) Scatter plot of $C_{dense}$ (AU) versus $C_{dilute}$ (AU). (**K**) Scatter plot of $C_{dense}$ (AU) and $C_{cytoplasm}$ (AU). (**L**) Boxplot showing $C_{dilute}$ (AU); p<0.001 by Kruskal Wallis Bonferroni treatment. (**M**) Scatter plot of $C_{dilute}$ (AU) versus $C_{cytoplasm}$ (AU). (**N**) Scatter plot of ΔΔG, J/mol versus $C_{dilute}$. (**O**) Boxplot of fold change in free energy of partition (ΔΔG, cal/mol); p<0.001; Kruskal Wallis Bonferroni treatment. (**P**) Representative images of A549 cells infected and mock infected with PR8 containing NP-WT and NP-Y289H treated and untreated with nucleozin for 1 hr (n=17–22). Scale bar = 10 μm. (**Q**) Boxplot of mean inclusion area per cell, per treatment p<0.001 by Kruskal Wallis Bonferroni treatment. (**R**) Boxplot of fold change in free energy of partition (ΔΔG, cal/mol) of viral inclusions arising in A549 cells infected and mock infected with PR8 containing NP-WT and NP-Y289H; p<0.001; Kruskal Wallis Bonferroni treatment.

The online version of this article includes the following figure supplement(s) for figure 4:

**Figure supplement 1.** Validation of method analysing thermodynamics parameters.

**Figure supplement 2.** Hardened inclusions are thermally stable.

inclusion molecular dynamics by fluorescence loss after photoactivation (FLAPh, *Figure 5D*). In a live imaging experiment, a region of interest (ROI) was photoactivated (*Figure 5E*), its decay profile was monitored for 120 s and the plot fitted to a single exponential model. At late points of infection, native levels of Rab11a versus overexpressed Rab11 conditions exhibited distinct vRNP decay profiles (*Figure 5F*), half-lives (14.48±2.2 s [mean ± SEM] and 52.15±9.6 s, respectively) and rate constants (0.06±0.04 s$^{-1}$ [mean ± SEM] and 0.02±0.003 s$^{-1}$, respectively), despite not changing the mobile and immobile fractions (*Figure 5G–I*, *Supplementary file 1* (Sheet 5) and *Figure 5—videos 7–12*). This indicates that although maintaining a liquid character, in conditions of overexpressing Rab11a, inclusions become less dynamic.

Next, we sought to assess if nucleozin altered the material properties of IAV inclusions. We first checked if nucleozin-treated viral inclusions maintained the ability to dissolve upon shock treatments, as illustrated in *Figure 6A*. We observed that native inclusions responded to shock treatment as expected, however, nucleozin strongly held inclusions together that did not dissolve when exposed to either hypotonic or 1,6-hexanediol shock treatments (*Figure 6B and C*, *Supplementary file 1* (Sheet 11)). This unresponsiveness to shock suggests that IAV inclusions undergo hardening when vRNP interactions are stronger.

Lastly, we measured the internal rearrangement in viral inclusions (*Figure 6P*). In native conditions, the photobleached region quickly disappeared, consistent with internal rearrangement of vRNPs inside IAV inclusions, whilst in nucleozin-treated inclusions, the photobleached area remained unaltered, revealing stiffness (several examples in *Figure 6Q* and *Figure 6—videos 9 and 10*).

To formally establish that IAV liquid inclusions can be hardened, we compared the dynamics of viral inclusions in the presence or absence of nucleozin using four different approaches. First, we assessed their movement and measured speed and displacement from their point of origin (*Figure 6D*). Native liquid inclusions (treated with sham vehicle – dimethyl sulfoxide [DMSO]) display a highly stochastic movement and long displacement, whilst nucleozin-hardened inclusions were less mobile with smaller displacement, as observed by analysing loss of movement in individual tracks (*Figure 6E*). There is an overall reduction in mean square displacement (MSD) with nucleozin (*Figure 6F*) that results in a lower MSD at 100 s (MSD$_{100 s}$ = 0.838 ± 1.17 μm$^2$ without nucleozin shifting to 0.057±0.22 μm$^2$ with treatment, median ± SD, *Figure 6F–G* and *Supplementary file 1* (Sheet 6)).

In a second approach, we measured the time that two droplets take to relax to a sphere upon fusion by coarsening assays (shifting the aspect ratio from 2 to 1, *Figure 6H*). DMSO-treated inclusions relax fast to a single sphere upon fusion (5.8±1.94 s; mean fusion time ± SEM), shifting the aspect ratio from 2 to 1. Nucleozin-treated inclusions retain a stable aspect ratio over time (*Figure 6I*), as they are

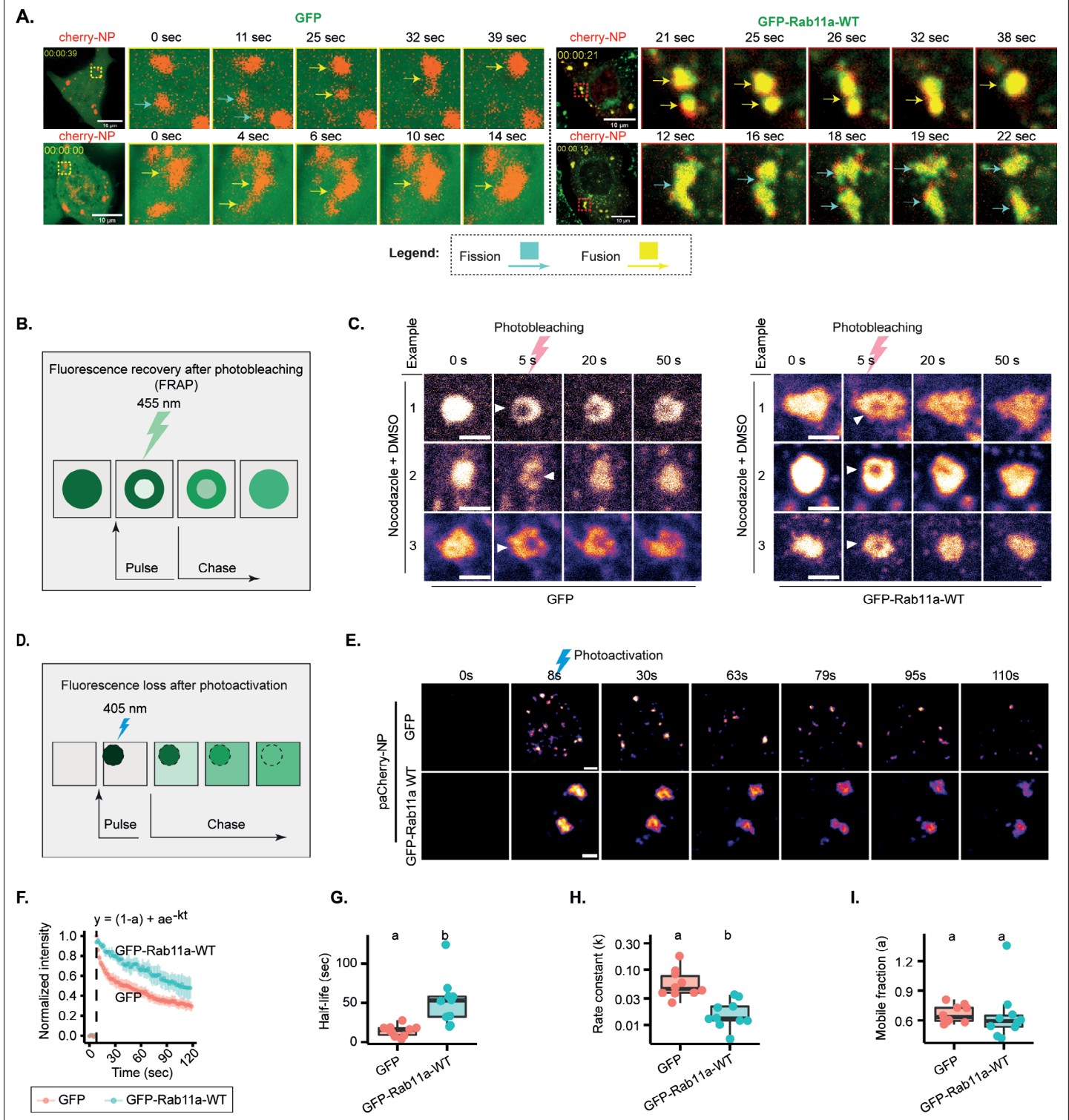

**Figure 5.** Changes in concentration of viral ribonucleoproteins (vRNPs) and Ras-related in brain 11a (Rab11a) modestly alter the material properties of viral inclusions. (A–I) A549 cells stably expressing GFP, or Rab11a-WT were transfected with a plasmid encoding mCherry-NP and simultaneously co-infected with PR8 virus at an MOI of 10 and were live imaged at 12–16 hr post-infection (hpi). (Number of cells (n)=10–18 for GFP and GFP-Rab11-WT.) (F–I) Each dot is the average value of measured parameters per cell. Above each boxplot, same letters indicate no significant difference between them, while different letters indicate a statistical significance at α=0.05 using one-way ANOVA, followed by Tukey multiple comparisons of means for parametric analysis, or Kruskal Wallis Bonferroni treatment for non-parametric analysis. All related values are displayed in *Supplementary file 1* (Sheet 5). (A) Representative time lapse images of fission (blue arrow) and fusion (yellow arrow) dynamics of viral inclusions in cells with endogenous levels or

*Figure 5 continued on next page*

*Figure 5 continued*

overexpressing Rab11a (*Figure 5—videos 1–4*). (**B**) Schematic depiction of an internal rearrangement of viral inclusion after an ROI within the inclusion is photobleached. (**C**) At 12 hpi, cells were treated with nocodazole (10 µg/mL) for 2 hr to reduce the highly stochastic motion of liquid influenza A virus (IAV) inclusions in GFP and GFP-Rab11a lines. Small regions inside IAV inclusions were photobleached to assess internal rearrangement of vRNPs (mCherry-NP as proxy). Time lapse pseudo-colour images show internal rearrangements after photobleaching (extracted from *Figure 5—videos 5; 6*). Scale bar = 2 µm. (**D**) Schematic of a fluorescence loss after photoactivation (FLAPh) experiment. (**E**) Time lapse pseudo-colour images showing the fluorescence loss in photoactivated IAV inclusions (photoactivatable paCherry-NP used as proxy) in GFP (n=10) or GFP-Rab11a cell lines (n=10) (extracted from *Figure 5—videos 7–12*). The analysis is a representative of two biological replicates. Bar = 2 µm. (**F**) Fluorescence intensity decay of photoactivated (paCherry-NP) normalised to GFP. Coloured lines are single exponential model fitting ($y_0 = (1-a) + ae^{-kt}$) of the data point, dots are the mean of the data per second, and vertical lines denote the standard deviation (SD) per time (**s**), (*Supplementary file 1* (Sheet 5)). (**G**) Half-life ($t_{1/2} = \frac{In(2)}{k}$, $k = rate\ constant$) of GFP and GFP-Rab11a developed inclusions decay post-activation (s); p=0.0003 by Kruskal Wallis Bonferroni treatment (*Supplementary file 1* (Sheet 5)). (**H**) Boxplot showing the rate constant, k, of liquid inclusions (using paCherry-NP as proxy) arising in GFP and GFP-Rab11a cell lines; p=0.0003 by Kruskal Wallis Bonferroni treatment (*Supplementary file 1* (Sheet 5)). (**I**) Boxplot showing the immobile fractions from FLAPh experiment calculated by the formula $y_0 = (1-a) + ae^{-kt}$, where 1-a is the immobile fraction arising from GFP and GFP-Rab11a cell lines; p=0.898 by Kruskal Wallis Bonferroni treatment (*Supplementary file 1* (Sheet 5)).

The online version of this article includes the following video(s) for figure 5:

**Figure 5—video 1.** Fusion and fission dynamics of influenza A virus (IAV) inclusions with endogenous Ras-related in brain 11a (Rab11a).
https://elifesciences.org/articles/85182/figures#fig5video1

**Figure 5—video 2.** Fission dynamics of influenza A virus (IAV) inclusions with endogenous Ras-related in brain 11a (Rab11a).
https://elifesciences.org/articles/85182/figures#fig5video2

**Figure 5—video 3.** Fusion dynamics of influenza A virus (IAV) inclusions overexpressing Ras-related in brain 11a (Rab11a).
https://elifesciences.org/articles/85182/figures#fig5video3

**Figure 5—video 4.** Fission dynamics of influenza A virus (IAV) inclusions overexpressing Ras-related in brain 11a (Rab11a).
https://elifesciences.org/articles/85182/figures#fig5video4

**Figure 5—video 5.** Photobleaching and observation of recovery after influenza A virus (IAV) inclusions in GFP stable expressing A549 lines.
https://elifesciences.org/articles/85182/figures#fig5video5

**Figure 5—video 6.** Photobleaching and observation of recovery after in influenza A virus (IAV) inclusions in GFP-Rab11a stable expressing cell lines.
https://elifesciences.org/articles/85182/figures#fig5video6

**Figure 5—video 7.** Fluorescence loss after photoactivation (FLAPh) in liquid inclusions showing paCherry-NP.
https://elifesciences.org/articles/85182/figures#fig5video7

**Figure 5—video 8.** Fluorescence loss after photoactivation (FLAPh) in liquid inclusions showing GFP.
https://elifesciences.org/articles/85182/figures#fig5video8

**Figure 5—video 9.** Fluorescence loss after photoactivation (FLAPh) in liquid inclusions showing the merged channels.
https://elifesciences.org/articles/85182/figures#fig5video9

**Figure 5—video 10.** Fluorescence loss after photoactivation (FLAPh) in liquid inclusions in cells overexpressing Ras-related in brain 11a (Rab11a) showing paCherry-NP.
https://elifesciences.org/articles/85182/figures#fig5video10

**Figure 5—video 11.** Fluorescence loss after photoactivation (FLAPh) in liquid inclusions in cells overexpressing Ras-related in brain 11a (Rab11a) showing GFP-Rab11a-WT.
https://elifesciences.org/articles/85182/figures#fig5video11

**Figure 5—video 12.** Fluorescence loss after photoactivation (FLAPh) in liquid inclusions in cells overexpressing Ras-related in brain 11a (Rab11a) showing the merged channels.
https://elifesciences.org/articles/85182/figures#fig5video12

unable to fuse (*Figure 6I–K*, *Supplementary file 1* (Sheet 7), *Figure 6—video 1* and *Figure 6—video 2*). The results demonstrate that nucleozin stiffens IAV inclusions.

In a third approach, inclusion molecular dynamics was tested by FLAPh (*Figure 6L*). In a live imaging experiment, an ROI was photoactivated (*Figure 6M*), its decay profile monitored for 120 s ,and the plot fitted to a single exponential model. DMSO- and nucleozin-treated inclusions exhibited distinct decay profiles (*Figure 6N*), with half-life of 14.41±0.9 s (mean ± SEM) and 85.02±19.8 s, respectively (*Figure 6O*, *Supplementary file 1* (Sheet 8) and *Figure 6—videos 3–8*). This indicates that nucleozin-treated inclusions become more static.

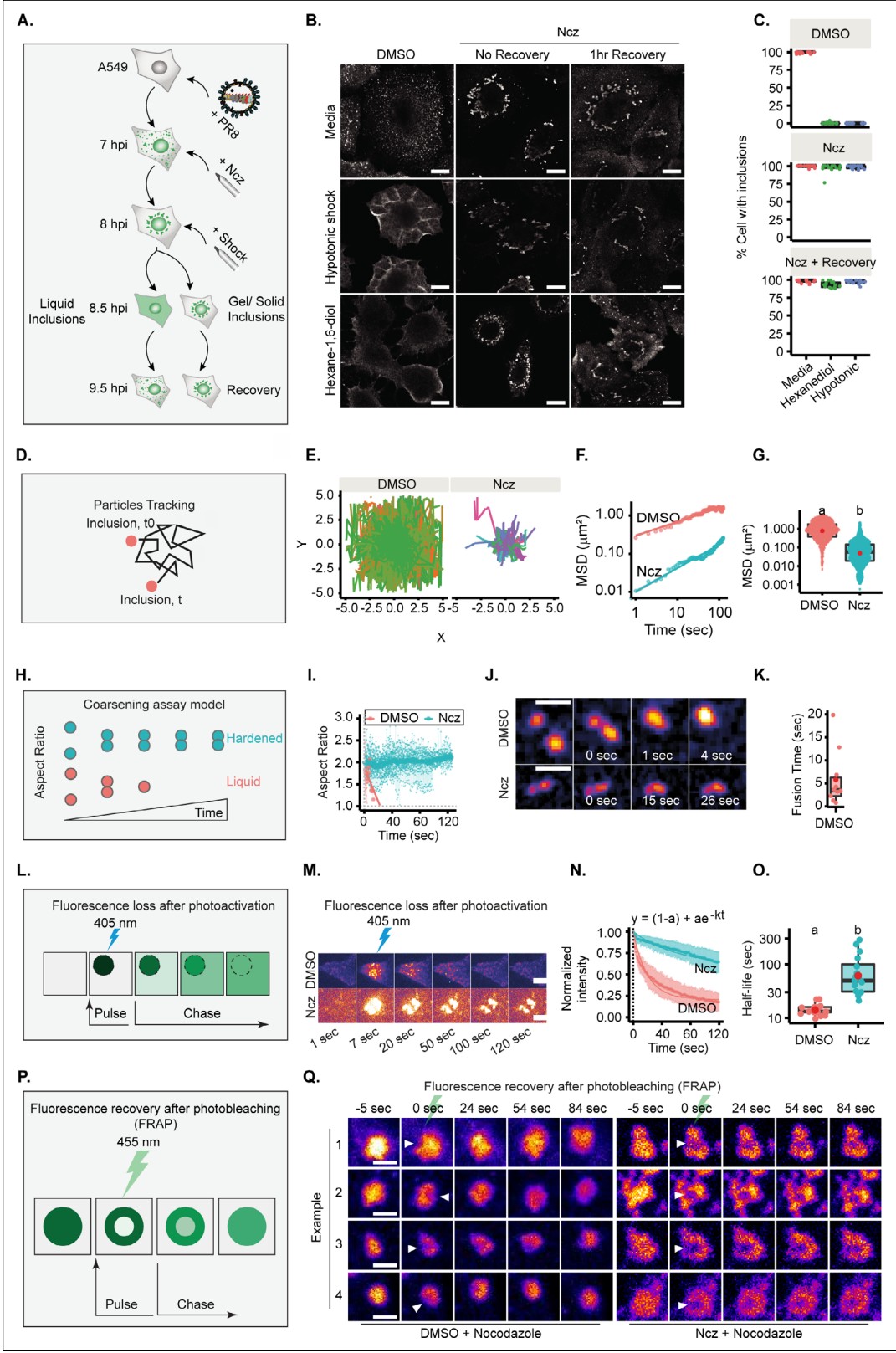

**Figure 6.** Increasing the strength/type of interactions between viral ribonucleoproteins (vRNPs) changes the material properties of liquid influenza A virus (IAV) inclusions. (**A–C**) A549 cells were infected at an MOI of 3 with PR8 virus and treated with 5 μM nucleozin (Ncz) or dimethyl sulfoxide (DMSO) at 7 hr post-infection (hpi). An hour later, cells were treated for 30 min with 80% water (hypotonic shock, Hyp), with 1,6-hexanediol (Hex) or

*Figure 6 continued on next page*

*Figure 6 continued*

complete media (CM) as control, before allowing recovery from stress treatment in CM for 1 hr. Cells were fixed, stained for NP for analysis by immunofluorescence, and the percentage of cells with IAV inclusions was scored manually. (D–K, P–Q) A549 cells were infected with PR8 virus at an MOI of 10 and simultaneously transfected with plasmids encoding (D–G) GFP-NP, (H–K) mcherry-NP, or (L–O) paGFP-NP and mcherry-NP. Cells were then live imaged after 12 hpi. (**A**) Experimental schematics of inclusion shock assay. (**B**) Representative images showing the response of IAV inclusions (nucleoprotein [NP], as proxy) to shock treatments after incubation in Ncz or DMSO. Scale bar = 10 μm. (**C**) Boxplot showing percentage cells with inclusions, after DMSO or Ncz treatment, by manual scoring; p<0.001 by Kruskal Wallis Bonferroni treatment. Analyses is a representative of three biological replicates. (Number of areas counted =15–17 and number of cells per area counted =39–56, *Supplementary file 1* (Sheet 11).) (**D**) Scheme showing how IAV inclusions were tracked over time. (**E**) Plot showing inclusion (GFP-NP, as proxy) particle trajectory when treated with DMSO (n=13) or Ncz (n=27). Data analysis was pooled from two biological replicates, *Supplementary file 1* (Sheet 6). (**F**) Graph showing the mean square displacement (μm$^2$) versus time (s) of IAV. (**G**) Boxplot depicting the resulting mean square displacement (μm$^2$) after 100 s tracking of IAV inclusions; p<0.001 by Kruskal Wallis Bonferroni treatment. (**H**) Schematics of the coarsening assay model, in which liquid and hardened IAV inclusions are represented by orange and blue dots, respectively. Unlike hardened inclusions, native liquid inclusions would fuse and relax to a spherical droplet. (**I**) Aspect ratio (AR) was used as a measure of IAV inclusion coalescence into a sphere (analysis is a pool of two biological replicates, n=10 for DMSO and 29 for Ncz). Horizontal grey dash lines depict a perfect sphere (aspect ratio =1). (**J**) Pseudo-coloured time lapse images of coalescing viral inclusions (GFP-NP used as proxy; extracted from *Figure 6—videos 1; 2*) in the presence or absence of Ncz. Scale bar = 2 μm. (**K**) Boxplot of the fusion time (s) of IAV liquid inclusions. Dots represent fusion time of individual fusion event (*Supplementary file 1* (Sheet 7)). (**L**) Schematic of a fluorescence loss after photoactivation (FLAPh) experiment. (**M**) Time lapse pseudo-colour images showing the fluorescence loss in photoactivated IAV inclusions (photoactivatable GFP-NP used as proxy) upon treatment with Ncz (n=16) or DMSO (n=17) (extracted from *Figure 6—videos 3–8*). The analysis is a representative of two biological replicates. Bar = 10μm. (**N**) Fluorescence intensity decay of photoactivated (paGFP-NP) normalised to the corresponding IAV inclusions expressing mcherry-NP. Coloured lines are single exponential model fitting (y$_0$ = (1-a) + ae$^{-kt}$) of the data point, dots are the mean of the data per second, and vertical lines denote the standard deviation (SD) per time (**s**), (*Supplementary file 1* (Sheet 8)). (**O**) Half-life (t$_{1/2}$ = $\frac{ln(2)}{k}$, $k = rate\ constant$) of liquid and hardened IAV inclusions decay post-activation (s); p=1.386e-6 by Kruskal Wallis Bonferroni treatment (*Supplementary file 1* (Sheet 8)). (**P**) Schematic depiction of an internal rearrangement of viral inclusion after an ROI within the inclusion is photobleached. (**Q**) A549 cells were transfected with plasmids encoding mcherry-NP and co-infected with PR8 virus at an MOI of 10. At 12 hr post-infection (hpi), cells were treated with nocodazole (10 μg/mL) for 2 hr to reduce the highly stochastic motion of liquid IAV inclusions and subsequently treated with DMSO or Ncz. Small regions inside IAV inclusions were photobleached to assess internal rearrangement of vRNPs (mCherry-NP as proxy). Time lapse pseudo-colour images show internal rearrangements after photobleaching (extracted from *Figure 6—videos 9; 10*). Scale bar = 10 μm.

The online version of this article includes the following video(s) for figure 6:

**Figure 6—video 1.** Coarsening assay in liquid inclusions.
https://elifesciences.org/articles/85182/figures#fig6video1

**Figure 6—video 2.** Coarsening assay in hardened inclusions.
https://elifesciences.org/articles/85182/figures#fig6video2

**Figure 6—video 3.** Fluorescence loss after photoactivation (FLAPh) in liquid inclusions showing cherry-NP.
https://elifesciences.org/articles/85182/figures#fig6video3

**Figure 6—video 4.** Fluorescence loss after photoactivation (FLAPh) in liquid inclusions showing paGFP-NP.
https://elifesciences.org/articles/85182/figures#fig6video4

**Figure 6—video 5.** Fluorescence loss after photoactivation (FLAPh) in liquid inclusions showing the merged channels.
https://elifesciences.org/articles/85182/figures#fig6video5

**Figure 6—video 6.** Fluorescence loss after photoactivation (FLAPh) in hardened inclusions showing cherry-NP.
https://elifesciences.org/articles/85182/figures#fig6video6

**Figure 6—video 7.** Fluorescence loss after photoactivation (FLAPh) in hardened inclusions showing paGFP-NP.
https://elifesciences.org/articles/85182/figures#fig6video7

**Figure 6—video 8.** Fluorescence loss after photoactivation (FLAPh) in hardened inclusions showing the merged channels.
https://elifesciences.org/articles/85182/figures#fig6video8

**Figure 6—video 9.** Photobleaching and observation of recovery after in liquid inclusions.
https://elifesciences.org/articles/85182/figures#fig6video9

*Figure 6 continued on next page*

**Figure 6—video 10.** Photobleaching and observation of recovery in hardened inclusions.
https://elifesciences.org/articles/85182/figures#fig6video10

---

Taken together, DMSO- and nucleozin-treated IAV inclusions exhibit distinct responses to shocks, dynamics, internal rearrangement, and coalescing properties, supporting that nucleozin hardens IAV liquid inclusions.

## Modifiers of strength/type of interactions between vRNPs hardens IAV liquid inclusions *in vivo*

Recently, the condensate-hardening drugs steroidal alkaloid cyclopamine and its chemical analogue A3 were shown to reduce viral titres in respiratory syncytial virus (RSV) infected mice (*Risso-Ballester et al., 2021*). However, at the organismal level, it was not demonstrated that RSV inclusion bodies in infected cells retained hardened features. To test if we could phenocopy the *in vitro* function of nucleozin, we aimed at analysing vRNP morphology inside the lung cells of infected mice. For this, we challenged mice with the IAV strain X31 for 2 days. At 30 min, 1 or 2 hr before the collection of the lungs, each mouse was treated with PBS (sham vehicle) or nucleozin, administered intranasally (*Figure 7A–C*). Interestingly, when we analysed viral inclusions under control conditions in cells of

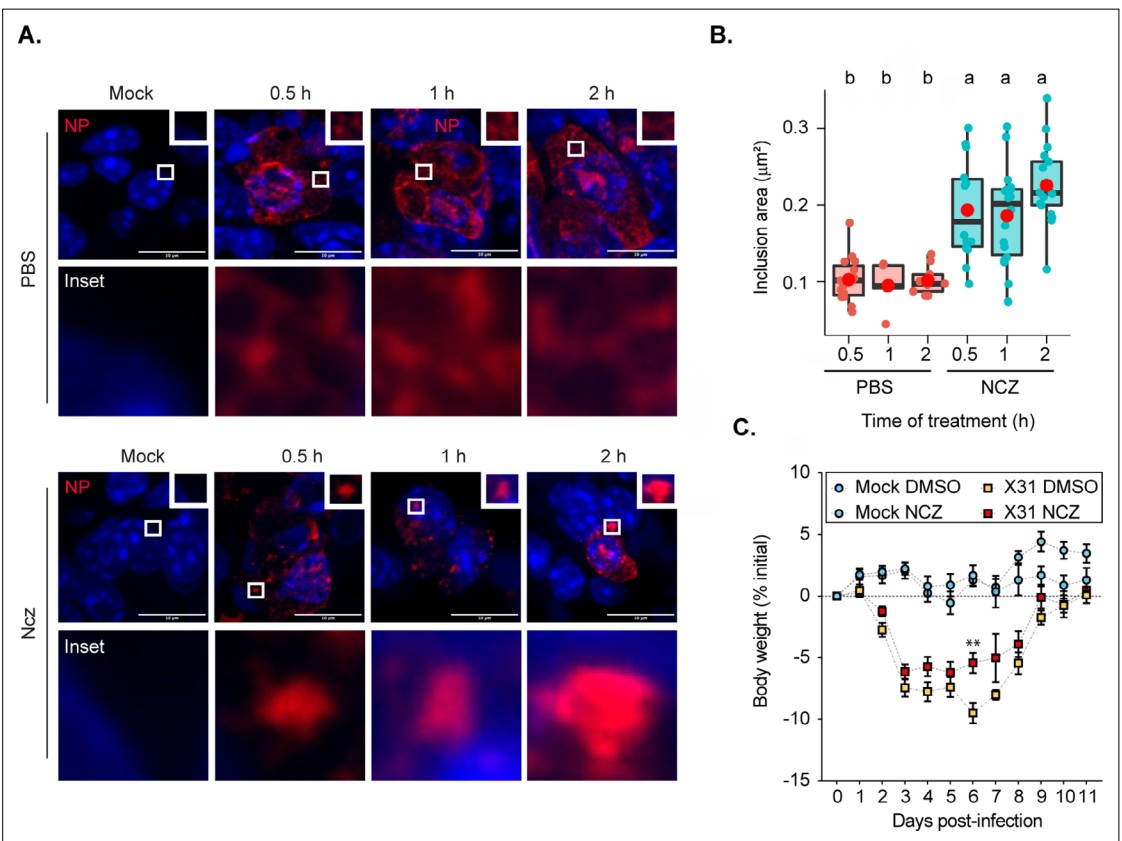

**Figure 7.** Hardened inclusions emerge *in vivo* when infected mice are treated with nucleozin. (**A–B**) Mice were intranasally infected with 4000 plaque forming units (PFU) of X31 virus, and after 2 days were intranasally administered PBS or 8.3 nmoles/g mice of nucleozin (Ncz) at 30 min, 1 or 2 hr before the collection of the lungs. Data were extracted from inclusions (nucleoprotein [NP], as proxy) from fixed immunofluorescence images of lung tissues (number of cells counted = 446–1694). (**A**) Representative immunofluorescence images show sections of lung tissue stained for NP (red) and nucleus (blue) after PBS or Ncz treatment. (**B**) Boxplot showing the mean area (µm²) of inclusions from cells in lung section; p=3.378e-8 by Kruskal Wallis Bonferroni treatment (*Supplementary file 1* (Sheet 9)). (**C**) Mice were pre-treated intraperitoneally with 8.3 nmoles/g mice Ncz or PBS for 1 hr before being intranasally infected with 1000 PFU of X31 virus, injected with a daily dose of Ncz or PBS for 11 days and the weight loss monitored daily. Body weight analysis is from a pool of two biological replicates. The number of mice is, in total, 23 for mock infected mice and 33 for infected mice (*Supplementary file 1* (Sheet 10)).

lungs of infected mice, we observed a punctate-like NP distribution. Upon nucleozin treatment, these cytosolic inclusions grew larger (inclusions per cell mean ± SEM: nucleozin 30 min, 0.101±0.006 $\mu m^2$; 2 hr, 0.226±0.012 $\mu m^2$, *Figure 7B*, *Supplementary file 1* (Sheet 9)). This indicates that the pharmacological induced modulator activity of nucleozin on liquid inclusions (*Kao et al., 2010*; *Amorim et al., 2013*) was retained *in vivo*. Having seen an effect in vRNP cytosolic localisation *in vivo*, we aimed at confirming a nucleozin-dependent abrogation of IAV infection in our system as reported before (*Kao et al., 2010*). In fact, nucleozin was reported to affect viral titres by 1 log and increase survival of IAV (A/Vietnam/1194/04 H5N1) infected mice by 50%. For this, we challenged nucleozin pre-treated mice with X31 and treated them with a daily dose of PBS (sham vehicle) or nucleozin. We found that nucleozin-treated mice had a faster recovery from viral infection (*Figure 7C*, *Supplementary file 1* (Sheet 10)). In sum, the data serves as proof of concept that the material properties of condensates may be targeted *in vivo*, in agreement with *Risso-Ballester et al., 2021*.

## Nucleozin rescues formation of hardened IAV inclusions in the absence of Rab11a

Given the possibility to harden IAV inclusions, it is important to define the molecular mechanisms conferring the liquid material properties of these condensates, which remain elusive. As Rab11a drives the formation of IAV inclusions (*Amorim et al., 2011*; *Eisfeld et al., 2011*; *Lakdawala et al., 2014*; *Vale-Costa et al., 2016*; *Alenquer et al., 2019*; *Veler et al., 2022*), we asked if nucleozin could artificially reform viral inclusions and mimic its behaviour in the absence of Rab11a. Stable cell lines expressing Rab11a dominant negative (DN) (henceforward Rab11a-DN) did not form IAV inclusions, as expected, maintaining vRNPs dispersed throughout the cytosol (*Figure 8A*). Interestingly, both Rab11a-WT and Rab11a-DN cell lines, in the presence of nucleozin, exhibited cytosolic puncta (despite smaller in Rab11a-DN lines, *Figure 8A–B*). This indicates that nucleozin bypasses the need for Rab11a to concentrate vRNPs, forming aberrant inclusions as predicted. We next tested the fusion ability of nucleozin-induced IAV inclusions in Rab11a-DN cell lines. Unlike native inclusions in WT cells, nucleozin-induced IAV inclusions in Rab11a-DN are not able to fuse in coarsening assays (*Figure 8C–E*). In sum, the liquid properties of IAV inclusions derived from flexible intersegment interactions and interaction with Rab11a harden to form stiff aggregates upon nucleozin treatment even when active Rab11a is absent.

## Nucleozin affects vRNP solubility in the absence of Rab11a without altering host proteome profile

Next, to understand how both the viral and host proteomes remodel in response to nucleozin treatment, we used a recently developed quantitative mass spectrometry-based approach called SPP (*Sridharan et al., 2019*). This is a lysate centrifugation assay, which can distinguish the soluble (supernatant) from insoluble (dense assemblies) protein pools. The majority of proteins annotated to be part of membraneless organelles, as well as many cytoskeletal proteins, exhibit prominent insolubility. In SPP, two aliquots of cellular lysates are extracted with either a strong (SDS) or a mild (NP40) detergent. Protein extracted with SDS represent the total proteome, while the supernatant of NP40-extracted lysate represents the soluble sub-pool. The ratio of NP40- and SDS-derived protein abundance represents the solubility of a protein (*Figure 9A*). Protein solubility is a proxy to track phase transition events in different cellular states. However, this measurement cannot distinguish between different events, such as solidification, phase separation, percolation, and gelation (*Alberti and Hyman, 2021*) that may underlie the phase transition.

To define the effect of nucleozin in viral inclusions, we compared proteome abundance and solubility profiles of Rab11a-DN cell lines, where the formation of liquid inclusion is blocked, with that of Rab11a-WT cell lines at 12 hpi, in the absence or presence of nucleozin (1 hr treatment) (*Figure 9A–E*, *Supplementary file 2* and *Supplementary file 3*). Nucleozin treatment did not induce significant alteration in host proteome abundance in both cell lines (*Figure 9B*). Similarly, no major changes in terms of protein solubility were observed for the host proteome during this treatment period (*Figure 9B*). Overall, our results suggest that nucleozin does not induce changes in cellular protein levels or their solubility.

In terms of the viral proteome, the abundance of all protein components of vRNPs (NP, PB1, PB2, PA) and M1 show a modest increase in Rab11a-DN cell lines (*Figure 9C*). On the solubility

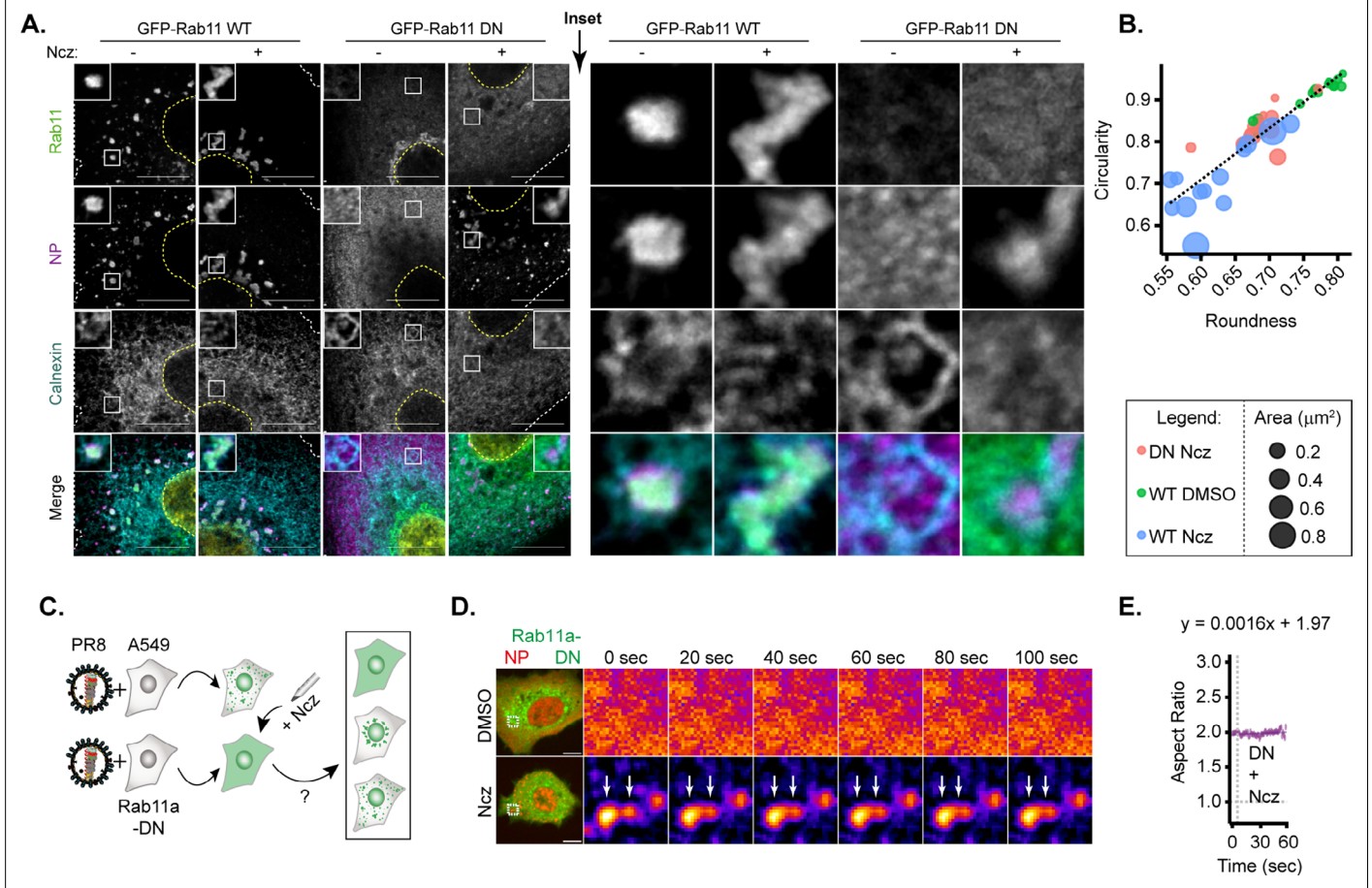

**Figure 8.** Only hardened inclusions emerge in nucleozin-treated Rab11a-DN cell line. (**A–B**) A549 cells constitutively expressing GFP-Rab11a-WT and GFP-Rab11a-DN were infected for 10 hr with PR8 at an MOI of 3 and treated with 5 µM nucleozin (Ncz) (n=10) or dimethyl sulfoxide (DMSO) (n=10) before fixing for analysis by immunofluorescence. The data are a representative analysis from biological duplicates. (**A**) Representative images of cells analysed by immunofluorescence staining using antibodies against viral protein nucleoprotein (NP) (magenta), host Rab11 (green), and ER (cyan). Nuclei and cell periphery delimited by yellow and white dashed line respectively, and white boxes are insets showing presence or absence of viral inclusions. Scale bar = 10 µm. (**B**) Scatter plot of circularity versus roundness of viral inclusions. (**C–E**) A549 cells constitutively expressing GFP-Rab11a-DN were transfected with mcherry-NP and co-infected with PR8 virus at an MOI of 3. At 12 hr post-infection (hpi), the cells were treated with 5 µM Ncz or DMSO for 10 min before imaging. (**C**) Schematic depicting the possible outcomes when Rab11a-DN cell lines are treated with Ncz. (**D**) Representative time lapse pseudo-colour images show fusion of IAV inclusions in a coarsening assay of PR8 infected Rab11a-DN cell line treated with Ncz or DMSO (extracted from *Figure 8—videos 1–6*). (**E**) Plot depicting the aspect ratio of fusing inclusions over time in infected Rab11a-DN cell line treated with Ncz.

The online version of this article includes the following video(s) for figure 8:

**Figure 8—video 1.** Coarsening assay of Rab11a-DN cells treated with dimethyl sulfoxide (DMSO) but lacking inclusions showing cherry-NP.
https://elifesciences.org/articles/85182/figures#fig8video1

**Figure 8—video 2.** Coarsening assay of Rab11a-DN cells treated with dimethyl sulfoxide (DMSO) but lacking inclusions showing paGFP-NP.
https://elifesciences.org/articles/85182/figures#fig8video2

**Figure 8—video 3.** Coarsening assay of Rab11a-DN cells treated with dimethyl sulfoxide (DMSO) but lacking inclusions showing the merged files.
https://elifesciences.org/articles/85182/figures#fig8video3

**Figure 8—video 4.** Coarsening assay of hardened viral inclusions formed by treating Rab11a-DN with nucleozin showing cherry-NP.
https://elifesciences.org/articles/85182/figures#fig8video4

**Figure 8—video 5.** Coarsening assay of hardened viral inclusions formed by treating Rab11a-DN with nucleozin showing paGFP-NP.
https://elifesciences.org/articles/85182/figures#fig8video5

**Figure 8—video 6.** Coarsening assay of hardened viral inclusions formed by treating Rab11a-DN with nucleozin showing the merged channels.
https://elifesciences.org/articles/85182/figures#fig8video6

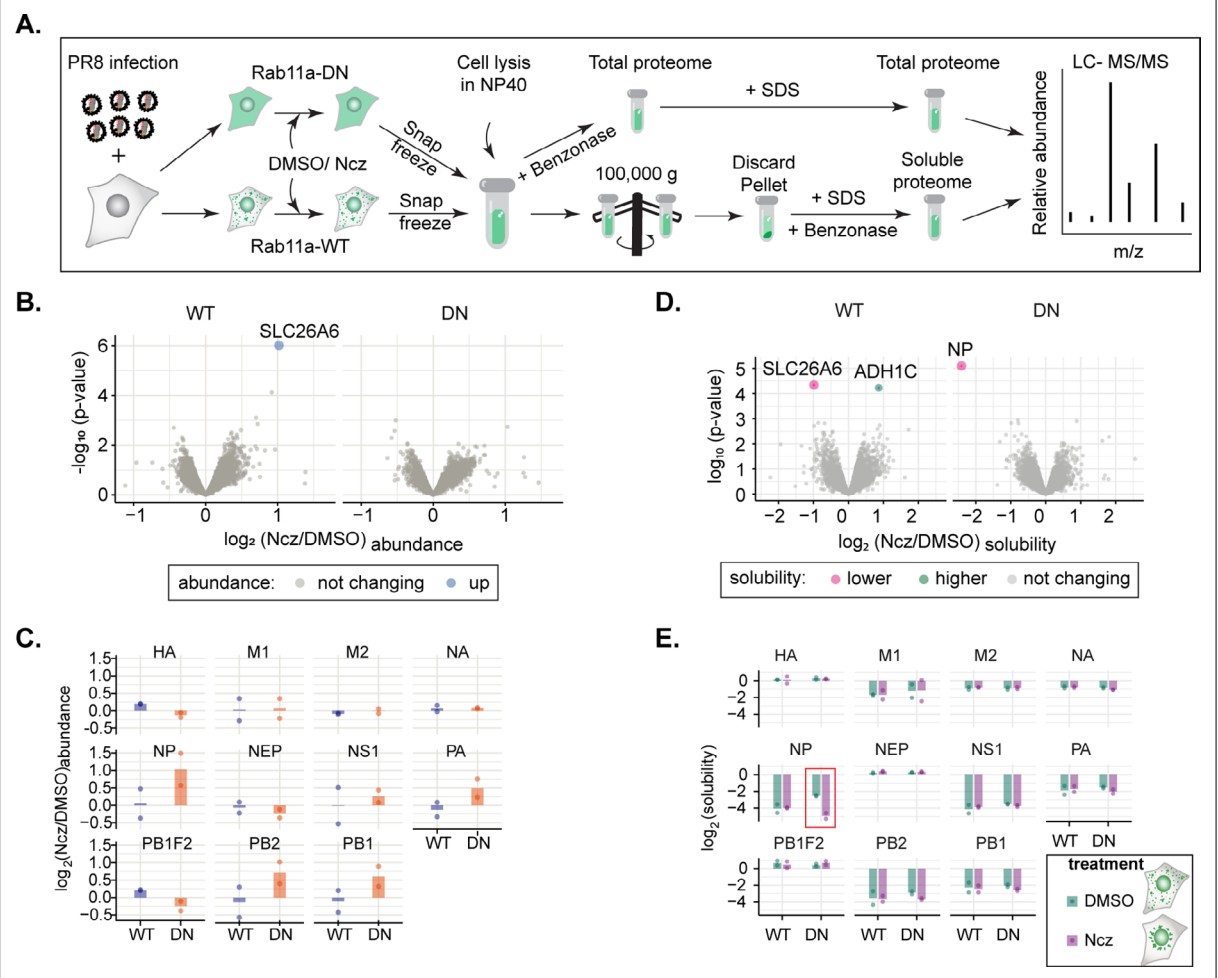

**Figure 9.** Hardening of influenza A virus (IAV) inclusions changes its proteome solubility. (A–E) A549 cells constitutively expressing GFP-Rab11a-WT or GFP-Rab11a-DN were infected for 12 hr with PR8 at an MOI of 5 and treated with 5 µM nucleozin (Ncz) or dimethyl sulfoxide (DMSO) for 1 hr. Thereafter, cells were lysed in mild (NP40) or strong detergent (SDS), while NP40 lysate was ultracentrifuged (100,000 × g) to pellet materials in condensates from the soluble fraction in the supernatant. Soluble and total host and viral proteome were identified by LC-MS/MS and solubility was determined as the ratio of soluble NP40- to SDS-derived total proteome abundances at the indicated timepoints. Data were from two biological replicates. (**A**) Schematic representation of solubility proteome profiling (SPP). (**B**) Volcano plot representing relative host protein abundance in Rab11a-WT and Rab11a-DN infected cell lines (at 12 hr post-infection [hpi]) after treatment with Ncz or DMSO. Differentially upregulated proteins in these conditions (statistical significance – see Materials and methods) are indicated in blue dots. (**C**) Bar graphic comparing abundances of viral proteins (in log$_2$ scale) in Rab11a-WT and Rab11a-DN cell lines PR8-infected (12 hpi) and treated with either Ncz or DMSO. (**D**) Volcano plot representing relative solubility of host and viral proteins in Rab11a-WT and Rab11a-DN infected cell lines (at 12 hpi) after treatment with Ncz. Differentially soluble proteins in these conditions (statistical significance – see Materials and methods) are indicated in pink and green dots. (**E**) Bar graph comparing solubility (in log$_2$ scale) of viral proteins when PR8 infected (12 hpi) Rab11a-WT and Rab11a-DN cell lines were treated with either Ncz or DMSO.

level, NP exhibited a prominent change. NP remains more soluble in Rab11a-DN lines compared to Rab11a-WT infected cells (fold change of 0.188, p=7.97e-6, *Figure 9C*). This corroborates the observation that vRNPs remain uniformly distributed in Rab11a-DN cells. Upon nucleozin treatment, SPP data reveal that the solubility of NP remains unaltered in Rab11a-WT cells, while increasing the proportion of NP in insoluble pool in Rab11a-DN cells (*Figure 9D–E*, red square). Although there were no changes in solubility by SPP, we observed IAV inclusions growing larger and hardening upon nucleozin treatment at the microscopic level in Rab11a-WT cells (*Figure 8A*). This can be explained, as vRNPs are already insoluble in viral inclusions before nucleozin treatment and the net increase in size of the inclusions does not result in higher insolubility of vRNPs. Both SPP and microscopy complement each other in the case of Rab11a-DN cells, as viral inclusions change from soluble to insoluble and become bigger upon nucleozin treatment. Overall, these

data substantiate our finding that vRNPs form Rab11a-dependent insoluble and liquid inclusions that undergo a distinctive (aberrant) phase transition upon nucleozin treatment.

PR8-infected Rab11a-WT and Rab11a-DN cells were treated with either DMSO (vehicle) or 5 μM of nucleozin for 1 hr. The protein abundance changes upon nucleozin (or DMSO) treatment in RAB11a-WT and Rab11a-DN cells is listed in this table.

## Discussion

In thermodynamics, the demixing from the surrounding media implies a preference of alike molecules to interact and self-sort, excluding the milieu. This is well understood for binary systems but deviate considerably for multi-component systems, even *in vitro* (*Klosin et al., 2020*; *Riback et al., 2020*; *Snead et al., 2022*). How living cells, that are complex multi-component systems at non-equilibrium, operate lacks understanding. Small alterations in the interactions, caused by changes in the environment or the interactome of the condensate, originate different self-assembled structures (*Riback et al., 2020*) that respond distinctly to thermodynamic variables such as concentration, temperature, and type/strength of interactions. For example, increasing the concentration in a system is mostly associated with more ordered, less flexible structures, however higher ordered structures were reported to arise in response to a concentration reduction (*Helmich et al., 2010*). Therefore, understanding how physical modulators of phase transitions impact the properties of condensates is key to comprehend how biological systems may be regulated (*Hermans et al., 2009*). IAV infection forms cytosolic liquid inclusions that are sites for genome assembly. Our study to address the fundamental question of whether the material properties of IAV inclusions may be modulated shows that IAV inclusions may be hardened by targeting vRNP interactions but not by lowering the temperature down to 4°C nor by altering the concentration of the factors that drive their formation. The data on temperature reveals that a decrease in the entropic contribution leads to a growth of condensates, as observed for other systems (*Falahati and Haji-Akbari, 2019*; *Hyman et al., 2014*; *Riback et al., 2020*), that is, however, mild and does not significantly impact the stability of the structures (*Figure 10*). Similarly, altering the concentration of drivers of IAV inclusions impact their size but not their material properties (*Figure 10*). This is unexpected because many studies have shown that changing the temperature or concentration of condensate drivers dramatically impacts their phase diagrams (*Bracha et al., 2018*; *Riback et al., 2020*; *Zhu et al., 2019*) and material properties (*Shin et al., 2017*). For influenza, these minor effects demonstrate that system is flexible, which may result from the necessity to maintain the liquid character over a wide range of vRNP concentration in the cytosol (low levels in the beginning and high at late stages of infection). The maintenance of the liquid character may be a regulated process involving fission and fusion events associated with the ER, as reported for other systems (*Lee et al., 2020*). In fact, IAV liquid inclusions develop in proximity to a particular part of a modified endoplasmic reticulum (ER) (*de Castro Martin et al., 2017*), the ER exit sites (*Alenquer et al., 2019*). In addition, the fusion and fission events of inclusions may be necessary to promote vRNP interactions, which is essential for genome assembly, as proposed before (*Eisfeld et al., 2015*; *Lakdawala et al., 2014*).

Defining the rules for hardening the condensates is important for understanding how biological condensates may be manipulated in cells and has consequences for development of novel antiviral treatments. By demonstrating that targeting the type/strength of interactions modulates the material properties of liquid viral inclusions in *in vitro* and *in vivo* models, we show that the development of molecules that affect the interactions between two components (such as post-translational modifications, local pH, or ionic strength or pharmaceutical modulators) should be prioritised over those increasing their concentration or local entropy (*Figure 9*). Such targeting may prevent off-target effects, especially by developing compounds able to distinguish free vRNP components from those in the supramolecular complex. In fact, the SPP herein reported demonstrates that it is possible to harden a liquid condensate without imposing changes in the host proteome abundance and solubility, which is important to increase specificity. However, a cost of targeting conserved molecules is the evolution of escape mutants (*Kao et al., 2010*; *Cheng et al., 2012*; *Hu et al., 2017*). Therefore, a concern to address in the future is how to design suitable combinatory therapies able to reduce their emergence. Since single nucleotide mutations underpin numerous resistance mechanisms to antivirals (*Lampejo, 2020*), an alternative is to engineer condensate hardening drugs that require multiple amino acid changes for escaping.

In this work, we explored the rules for hardening IAV liquid condensates. Other alternatives to modulate the material properties tailored for function can be developed. For example, accumulating evidence shows that blocking viral inclusion formation hinders viral infection (*Amorim et al., 2011*; *Eisfeld et al., 2011*; *Momose et al., 2011*; *Vale-Costa et al., 2016*; *de Castro Martin et al., 2017*; *Vale-Costa and Amorim, 2017*; *Alenquer et al., 2019*; *Han et al., 2021*; *Veler et al., 2022*). Herein, we observe that increase in temperature biases the system to dissolving viral inclusions (that is complete at 43.5°C), therefore activating exothermic reactions close to IAV inclusions may lead to their dissolution. Furthermore, it has been previously demonstrated that blocking Rab11 pathway, directly or indirectly, hampers viral infection (*Amorim et al., 2011*; *Eisfeld et al., 2011*; *Momose et al., 2011*; *Han et al., 2021*). Future research could also explore this route. As Rab11a has emerged as a key factor for the replication of members of many unrelated viral families relevant for human health (Bunyaviridae, Filoviridae, Orthomyxoviridae, Paramyxoviridae, and Pneumoviridae), targeting its activity may serve as a pan-antiviral strategy (*Amorim et al., 2011*; *Bruce et al., 2010*; *Nakatsu et al., 2013*; *Nanbo and Ohba, 2018*; *Cosentino et al., 2022*).

## Limitations of the study

Understanding condensate biology in living cells is physiological relevant but complex because the systems are heterotypic and away from equilibria. This is especially challenging for influenza A liquid inclusions that are formed by eight different vRNP complexes, which although sharing the same structure, vary in length, valency, and RNA sequence. In addition, liquid inclusions result from an incompletely understood interactome where vRNPs engage in multiple and distinct intersegment interactions bridging cognate vRNP-Rab11 units on flexible membranes (*Chou et al., 2013*; *Gavazzi et al., 2013*; *Sugita et al., 2013*; *Shafiuddin and Boon, 2019*; *Haralampiev et al., 2020*; *Le Sage et al., 2020*). At present, we lack an *in vitro* reconstitution system to understand the underlying mechanism governing demixing of vRNP-Rab11a-host membranes from the cytosol. This *in vitro* system would be useful to explore how the different segments independently modulate the material properties of inclusions, explore if condensates are sites of IAV genome assembly, determine thermodynamic values, thresholds accurately, perform rheological measurements for viscosity and elasticity, and validate our findings. The results could be compared to those obtained in cell systems to derive thermodynamic principles happening in a complex system away from equilibrium. Using cells to map how liquid inclusions respond to different perturbations provide the answer of how the system adapts *in vivo*, but has limitations. One of the constraints of using cells in this work relates to the range and precision of the concentrations we can vary in our system. Herein, we compared endogenous Rab11a cellular levels to a single pool of transduced cells that contained low, but still heterogeneous, levels of Rab11a as a way to avoid toxicity and/or uncharacterised effects of exceedingly high concentration of Rab11a in the cell. To minimise this limitation, we combined overexpressing Rab11a with a range of low and high levels of vRNPs (analysing the entire time course of infection) to understand if a combination of high levels of vRNPs and of Rab11a could synergistically change the material properties of IAV inclusions. Technically, we retrieved thermodynamic parameters (such as $C_{dense}$, $C_{dilute}$, shape, size) from images in which vRNPs were stained using antibodies. As mentioned above, antibodies may have some difficulty accessing the inner of the condensates, which could affect measurements of $C_{dense}$ or the total concentration of vRNPs in the cytosol. This could, in turn, affect the calculation of some thermodynamic parameters, including nucleation density and Gibbs free energy. Importantly, differences in antibody access may depend on some condensate properties, which may even change during infection, inducing artefactual trends. Alternatives to using antibodies comprise viruses with fluorescently tagged vRNPs, like the reported virus with the viral polymerase PA tagged with GFP (*Bhagwat et al., 2018*). This virus could be the ideal approach to evaluate inclusion thermodynamics for the whole study, were it not for the fact that the virus is attenuated, exhibiting delayed infection, reduced levels of viral proteins and of accumulation of vRNPs in the cytosol, with viral inclusions forming later in infection. As such, the use of these viruses would also affect the thermodynamic analyses. Future assessment of when one approach is more suitable over the other is needed. A second technical limitation relates that we acquired data from images in z-stacks as the sum of slices at specific snapshots of infection. However, although requiring a very complex imaging analysis that we lack, in the ideal scenario, the analysis should have been done using the whole

volumetry of each viral inclusion, and using live images quantified over time that is yet to be reported.

# Materials and methods

## Key resources table

| Reagent type (species) or resource | Designation | Source or reference | Identifiers | Additional information |
|---|---|---|---|---|
| Cell line (*Homo sapiens*) | A549 | ATCC | CCL-185 | Human alvelolar basal cell |
| Cell line (*Canis familiaris*) | MDCK.1 | ATCC | CRL-2935 | Mardin-Darby canine kidney cell |
| Antibody | Anti-Rab11a (Rabbit polyclonal) | Proteintech | Cat# 15903-1-AP, RRID:AB_2173458 | IF(1:100) |
| Antibody | Anti-Calnexin (Rabbit polyclonal) | Abcam | Cat#22595; RRID:AB_2069006 | IF(1:1000) |
| Antibody | Anti-NP (Mouse monoclonal) | Abcam | Cat#20343; RRID: AB_445525 | IF(1:1000) |
| Recombinant DNA reagent | GFP-NP (plasmid) | *Amorim et al., 2011* | N/A | GFP version of NP |
| Recombinant DNA reagent | mCherry-NP (plasmid) | *Amorim et al., 2011* | N/A | Cherry version of NP |
| Recombinant DNA reagent | paCherry-NP (plasmid) | This paper | N/A | Cherry photoactivatable version of NP |
| Recombinant DNA reagent | paGFP-NP (plasmid) | This paper | N/A | GFP photoactivatable version of NP |
| Sequence-based reagent | Gipc1_F | This paper | PCR primers | GGGAAAGGACAAAAGGAACCC |
| Sequence-based reagent | Gipc1_R | This paper | PCR primers | CAGGGCATTTGCACCCCATGCC |
| Sequence-based reagent | paGFP_L64F/T65S_Fw | This paper | PCR primers | CCCTCGTGACCACCTTCAGCTACGGCGTGCAGT |
| Sequence-based reagent | paGFP_T203H/A206K_Fw | This paper | PCR primers | GACAACCACTACCTGAGCCACCAGTCCAAGCTGA GCAAAGACCCCAAC |
| Sequence-based reagent | paGFP_V163A_Fw | This paper | PCR primers | GAAGAACGGCATCAAGGCGAACTTCAAGATCCGCC |
| Sequence-based reagent | paCherry_NP_NheI_Fw | This paper | PCR primers | GATCCGCTAGCGGTCGCCACCATGG |
| Sequence-based reagent | paCherry_NP_XhoI_Rv | This paper | PCR primers | GCGCCTCGAGGATCTGAGTCCGGACTTGTA |
| Chemical compound, drug | DMEM, high glucose, pyruvate, no glutamine (Gibco) | Thermo Fisher | Cat# 21969035 | |
| Chemical compound, drug | L-Glutamine | Thermo Fisher | Cat# 25030024 | |
| Chemical compound, drug | OPTIMEM-I W/GLUTAMAX-I (CE) | Thermo Fisher | Cat# 51985026 | |
| Chemical compound, drug | Leibovitz's L-15 Medium, no phenol red (Gibco) | Life Technologies | Cat# 21083–027 | |
| Chemical compound, drug | Lipofectamine LTX Reagent with PLUS Reagent (Invitrogen) | Thermo Fisher | Cat# 15338100 | |
| Chemical compound, drug | Penicillin-Streptomycin Solution | Biowest | Cat# L0022-100 | |
| Chemical compound, drug | Dimethyl sulfoxide (DMSO) | BioLabs | Cat# B0515A | |

*Continued on next page*

*Continued*

| Reagent type (species) or resource | Designation | Source or reference | Identifiers | Additional information |
|---|---|---|---|---|
| Chemical compound, drug | Formaldehyde, extra pure, solution 37–41%, AR grade (Fisher Chemical) | Acros | Cat# 10231622 | |
| Chemical compound, drug | Fetal Bovine Serum, qualified, heat inactivated, Brazil (Gibco) | Thermo Fisher | Cat# 10500064 | |
| Chemical compound, drug | Nucleozin | Target Mol | Cat# 282T7330 | |
| Chemical compound, drug | 1,6-Hexanediol | Aldrich | Cat# 240117–50G | |
| Chemical compound, drug | Triton X-100 | Sigma | Cat# X100 | |
| Chemical compound, drug | Dako Faramount Aqueous Mounting Medium | Agilent Technologies | Cat# S3025 | |
| Chemical compound, drug | Complete protease inhibitor cocktail | Merck | Cat# 11836170001 | |
| Chemical compound, drug | PhosphoStop | Merck | Cat# 4906837001 | |
| Chemical compound, drug | RNasin Plus RNase Inhibitor | Promega | Cat# N2615 | |
| Chemical compound, drug | NP-40 | Thermo Fisher Scientific | Cat# FNN0021 | |
| Chemical compound, drug | SDS | NZYTech | Cat# MB01501 | |
| Chemical compound, drug | Benzonase Nuclease HC | Merck | Cat# 71206-3 | |
| Chemical compound, drug | Ethanol | VWR Chemicals | Cat# 20821.330 | |
| Chemical compound, drug | Sequencing Grade Modified Trypsin | Promega | Cat# V5111 | |
| Chemical compound, drug | Lysyl Endopeptidase, Mass Spectrometry | Wako | Cat# 125-05061 | |
| Chemical compound, drug | HEPES | Alfa Aesar | Cat# A14777 | |
| Chemical compound, drug | TMT-16plex reagents | Thermo | Cat# A44522 | |
| Chemical compound, drug | PIERCE BCA protein assay | Thermo | Cat# 23225 | |
| Software, algorithm | FIJI | ImageJ | RRID:SCR_002285 | https://imagej.net/software/fiji/ |
| Software, algorithm | R Project for Statistical Computing | R | RRID:SCR_001905 | https://www.r-project.org/ |
| Software, algorithm | Trackmate plugin | *Tinevez et al., 2017*; *Ershov et al., 2022* | RRID:SCR_002285 | https://imagej.net/plugins/trackmate/ |
| Software, algorithm | *limma* | *Ritchie et al., 2015*. | RRID:SCR_010943 | https://bioconductor.org/packages/release/bioc/html/limma.html |
| Software, algorithm | ClusterProfiler (R Bioconductor) | *Yu et al., 2012*. | RRID:SCR_016884 | https://bioconductor.org/packages/release/bioc/html/clusterProfiler.html |
| Software, algorithm | isobarQuant | Quant | https://doi.org/doi:10.18129/B9.bioc.isobar | https://www.bioconductor.org/packages/release/bioc/html/isobar.html |
| Software, algorithm | Mascot 2.4 (Matrix Science) | Mascot | RRID:SCR_014322 | http://www.matrixscience.com/server.html |

| Reagent type (species) or resource | Designation | Source or reference | Identifiers | Additional information |
|---|---|---|---|---|
| Other | Hoechst stain | Thermo Fisher Scientific | H3570 | (1 µg/mL) |

## Resources availability

### Lead contact

Further information and requests for resources and reagents should be directed to and will be fulfilled by the lead contact, Maria Joao Amorim (mjamorim@igc.gulbenkian.pt, mjamorim@ucp.pt).

### Materials availability

Materials produced under this study, including the photo-activatable GFP-NP (paGFP-NP) and photo-activatable mCherry-NP (paCherry-NP), may be requested to the corresponding author. Materials are subjected to MTA agreement to acknowledge the work of the authors.

### Cell lines

GFP-Rab11a-WT and GFP-Rab11a-DN cell lines were produced in-house and characterised in *Vale-Costa et al., 2016*, from the human alvelolar basal cell (A549, ATCC (CCL-185)), that were also used in this study. These cells were authenticated by ATCC with STR Profiling Results (StRC3941). Madin-Darby Canine Kidney (MDCK-I, acquired from ATCC) were used for their phenotypic properties (plaque assays) as an assay tool to titrate virus, but not the subject of the study themselves. MDCKs were not authenticated. A549s and MDCKs were obtained from Prof. Paul Digard, Roslin Institute, UK, as part of collaborative work. All cells are routinely checked for mycoplasma contamination and tested negative. Cells are routinely checked for mycoplasma contamination and tested negative. Cells were cultured in Dulbecco's Modified Eagle Medium (DMEM) supplemented with 10% fetal bovine serum (FBS), 2 mM L-glutamine, and 1% (v/v) penicillin-streptomycin. GFP-Rab11a cell lines were cultured/maintained in DMEM supplemented with 1.25 µg/mL puromycin. Cells were maintained in a humidified incubator at 37°C and 5% v/v atmospheric $CO_2$.

### Viruses

Reverse-genetics engineered A/Puerto Rico/8/34 virus (PR8 WT; H1N1) (*de Wit et al., 2007*) wild-type or carrying the mutation in NP Y289H as described in *Kao et al., 2010*, was used to infect all cell types and titrated by plaque assay in MDCK cells, while X31 virus (a reassortant virus carrying HA and NA segments from A/Hong-Kong/1/1968 (H3N2) [*Matrosovich et al., 2007*] in the background of PR8) was used to infect mice. Infection for live imaging were done at an MOI of 10, with viral infections for immunofluorescence at an MOI of 3 or 5.

### Plasmids construct

Plasmids used in the study and their source are found in Key resources table. Two plasmids were created for this work: paGFP-NP and paCherry-NP.

The paGFP-NP was generated based on the GFP-NP plasmid. Mutations as described in *Patterson and Lippincott-Schwartz, 2002*, were directly introduced using oligonucleotide primers and the QuikChange Multi Site-Directed Mutagenesis Kit (Agilent). The following primers were used:

paGFP_L64F/T65S_Fw:
5'- CCCTCGTGACCACCTTCAGCTACGGCGTGCAGT -3'
paGFP_T203H/A206K_Fw:
5'-GACAACCACTACCTGAGCCACCAGTCCAAGCTGAGCAAAGACCCCAAC -3'
paGFP_V163A_Fw:
5'- GAAGAACGGCATCAAGGCGAACTTCAAGATCCGCC -3'

To construct paCherry-NP, paCherry was amplified from pPAmCherry1-C1 (pPAmCherry1-C1, which was a gift from Vladislav Verkhusha [Addgene plasmid # 31929; http://n2t.net/addgene:31929; RRID:Addgene_31929, *Subach et al., 2009*]). The subsequent PCR was subcloned into pEGFP-NP plasmid (without GFP) using NheI and XhoI restriction sites. The following primers were used:

paCherry_NP_NheI_Fw:
5'- GATCCGCTAGCGGTCGCCACCATGG -3'
paCherry_NP_XhoI_Rv:
5'- GCGCCTCGAGGATCTGAGTCCGGACTTGTA -3'

## Animals

All experiments involving mice were performed using 8-week-old littermate C57BL/6J, female mice grouped by random sampling under specific pathogen-free conditions at the Instituto Gulbenkian de Ciência (IGC) biosafety level 2 animal facility (BSL-2).

## Mice infection

Female C57Bl/6 mice were infected with 1000 pfu of X31 (A/X31; H3N2) virus, 1 hr after being intra-peritoneally injected with DMSO or 450 µg of nucleozin. In the following 6 days, mice were injected daily with DMSO or 450 µg of nucleozin. At days 3 and 6 post-infection, mice were sacrificed, and lungs were collected to determine viral titres by plaque assays (using MDCK infected with a set of serial dilutions from the homogenised lung tissue samples). Mice were daily weighed for bodyweight assessment during all the course of infection (until 11 dpi).

For *in vivo* analysis of viral inclusions, female C57Bl/6 mice were infected with 4000 pfu of X31 (A/X31; H3N2) virus for 2 days. At 30 min, 1 hr, or 2 hr before the collection of the lungs, each mouse was intranasally treated with DMSO (vehicle) or 69 µg of nucleozin. Then, lungs were collected to determine viral titres by plaque assays (for controlling infection) and for histology processing. All animals were included in data analysis except animals that when being infected did not inhale the total 30 µL of PBS containing viruses (four mice), and four animals that crossed the 20% weight limit and were euthanised. Processing for the immunofluorescence of lung slices is under the Microscopy and image processing.

## Plaque assay

For viral titre measurement, A549 cells were seeded for 24 hr, infected at MOI of 3 in DMEM supplemented with 2 mM L-glutamine and 1% (v/v) penicillin/streptomycin and devoid of sera for 45 min at 37°C and 5% $CO_2$. The supernatants were subjected to a plaque assay in MDCK cells to calculate the virus titres, as described previously (*Matrosovich et al., 2006*).

## Drug treatment

Nucleozin was dissolved in DMSO and used at a final concentration of 2 µM (immunofluorescence staining and virus titres) or 5 µM (live imaging), while 1,6-hexanediol was dissolved in DMEM and used at 5% (w/v).

## Microscopy and image processing

For immunofluorescence, A549 cells were fixed in 4% (v/v) paraformaldehyde for 10 min and permeabilised with Triton X-100 (0.2% (v/v)), incubated in primary antibodies for 1 hr at room temperature (RT), washed (3×) in PBS/1% FBS (v/v) and finally incubated in Hoechst and Alexa Fluor conjugated secondary antibodies for 45 min at RT. Antibodies used were rabbit polyclonal against Rab11a (1:100; Proteintech, 15903-1-AP) and calnexin (1:1000, Abcam, 22595). Antibody against NP was mouse monoclonal (1:1000; Abcam, 20343). Secondary antibodies were all from the Alexa Fluor range (1:1000; Life Technologies). Following washing in PBS, cells were mounted with Dako Faramount Aqueous Mounting Medium and single optical sections were imaged with a Leica SP5 live or stellaris confocal microscope using the photon counter mode.

For immunofluorescence of lung slices, heat-induced epitope retrieval was performed with citrate buffer using a microwave in lung slices from mock and infected mice. The lung slices were then washed with PBS and blocked with a solution containing PBS, 2%BSA (w/v), and 0.5%Triton (v/v) in a humidified chamber for 1 hr at RT and 1 hr at 4°C. Subsequently, lung slices were washed twice with PBS and incubated with a rabbit polyclonal primary antibody against NP (1:1000) for 1 hr at RT, followed by a two-time wash in PBS, before a 30 min incubation in Hoechst and Alexa Fluor conjugated secondary antibody (1:500) at RT. Finally, lung slices were washed twice with PBS and dipped in water. Slices were mounted with Dako Faramount Aqueous Mounting Medium (Dako) and single optical sections

and z-stacks image were imaged with a Leica SP5 live confocal microscope using the photon counter mode. Lung immunofluorescence analysis were done blindly.

Using the function sum of slices, all z-stacked images were projected to 2D and inclusion and its cytoplasmic milieu were segmented and analysed using Lab-custom ImageJ macros and R analytics scripts.

## Determining inclusion topology and thermodynamics

Data was acquired from independent biological replicates of at least two experiments. All data, including sample size, mean, standard error of mean, median, standard deviation, tailored to each experiment can be found in *Supplementary file 1*. To assess the shape of liquid inclusions, we used ImageJ plugins to measure a series of parameters with the following formulas: Circularity = 4π × Area/(Perimeter)$^2$ with a value of 1 indicating a perfect circle. As the value approaches 0, it indicates an increasingly elongated shape. Uses the heading Circ (circularity); Aspect ratio=Major Axis/Minor axis. The aspect ratio of the particle's fitted ellipse. Uses the heading AR; Roundness = 4 × Area/(π × (Major axis)$^2$) or the inverse of Aspect Ratio. Uses the heading Round; Solidity = Area/Convex Area. To determine the total concentration of vRNPs (NP as proxy) transported to the cytoplasm in relation to vRNPs produced in the nucleus ($\frac{Ccytoplasm}{Cnucleus}$), a sum of slices of z-stacked images were used, otherwise, single plane images were analysed for other parameters. We used a custom (Fiji Is Just) ImageJ 2.1.0/1.53p script for image processing using the following pipeline: (1) Segment cell periphery. (2) Segment and remove nucleus from the cell to make the cytoplasm. (3) From the cytoplasm, segment inclusions. (4) Analyse the cytoplasm, nucleus, and inclusions for number and topological shape descriptors. (5) Using the appropriate segmented region, measure the MFI (as proxy of concentration) of cell, nucleus, cytoplasm, and cytoplasmic inclusion (see *Figure 1B*).

Using the method published by *Riback et al., 2020*, as template, we determined C$_{dense}$ as the MFI of the segmented inclusion while C$_{dilute}$ was extrapolated from remaining cytoplasmic vRNP intensity outside the inclusions. We picked the best approach out of three to measure C$_{dilute}$. (1) Use ROIs from randomly selected cytoplasmic areas lacking inclusions. The limitation with this method is that inclusions are highly abundant in the cytoplasm of infected cells and are nearly impossible to manually or automatically draw without selecting regions containing inclusions. (2) Use an enlarged ROI band around the inclusions. This was easy to automate but limited by the overlap with other ROI bands due to the density of IAV inclusions in the infected cell. (3) Use ROI of the entire cytoplasm devoid of viral inclusions. This was easy to automate, lacks overlap with other ROIs and serves as the cleanest strategy when compared to strategy 2 (*Figure 2—figure supplement 1*, *Figure 3—figure supplement 1*, *Figure 4—figure supplement 1A–H*). We used strategy 3 to determine the C$_{dilute}$.

Partition coefficient (K) and free energy (ΔG) were derived based on *Riback et al., 2020*, publication; where K = $\frac{cdense}{cdilute}$ and ΔG = -RTlnK. Inclusion saturation concentration (C$_{sat}$) is the threshold C$_{dilute}$ where inclusion begins to appear (~6 hpi) and is calculated as the minimum C$_{dilute}$ in cells with observable viral inclusions. The change in free energy was normalised to 3 hpi an infection stage with nuclear vRNP staining lacking cytoplasmic inclusions and was represented as ΔΔG = ΔG – ΔG$_{(3 hpi)}$.

## Live imaging, photoactivation, photobleaching

All experiments resulted from at least two biological replicates. A549 cells were seeded in eight-well glass-bottomed dish (Ibidi) and grown overnight in OptiMEM (37°C, 5% CO$_2$). Cells infected with PR8 at an MOI of 10 were transfected simultaneously with 200 ng/μL GFP-NP or cherry-NP plasmid and/or either paCherry-NP or paGFP-NP using lipofectamine LTX . Cells were imaged using OptiMEM or Leibovitz medium with a 63× oil immersion Nikon objective (NA = 1.4) on Roper TIRF, AiryScan, or spinning disk confocal (SoRa) microscopes equipped with temperature (37°C) and CO$_2$ (5%) regulated chamber and stage. Inclusions at a specified ROI was activated by blue light (405 nm laser) at 100% intensity and imaged at 1 frame/s for 2 min using 488 and 568 nm lasers for GFP and cherry, respectively. Photoactivation data were post-processed in FIJI (Image J) using a modified FLAPh algorithm and analysed with a lab-custom R script. Model was obtained using single exponential curve fitting. y = (1-a) + ae$^{-kt}$, a=mobile fraction, K=decay rate constant (per second, s$^{-1}$), t=time (s).

For image-based pulse-chase photobleaching, cells were transfected with 250 ng of GFP-NP and immediately superinfected with PR8 at an MOI of 10. At 12 hpi, media was substituted for Leibovitz L-15 media to buffer CO$_2$ and data acquisition started on Zeiss AiryScan LSM 980 microscope with

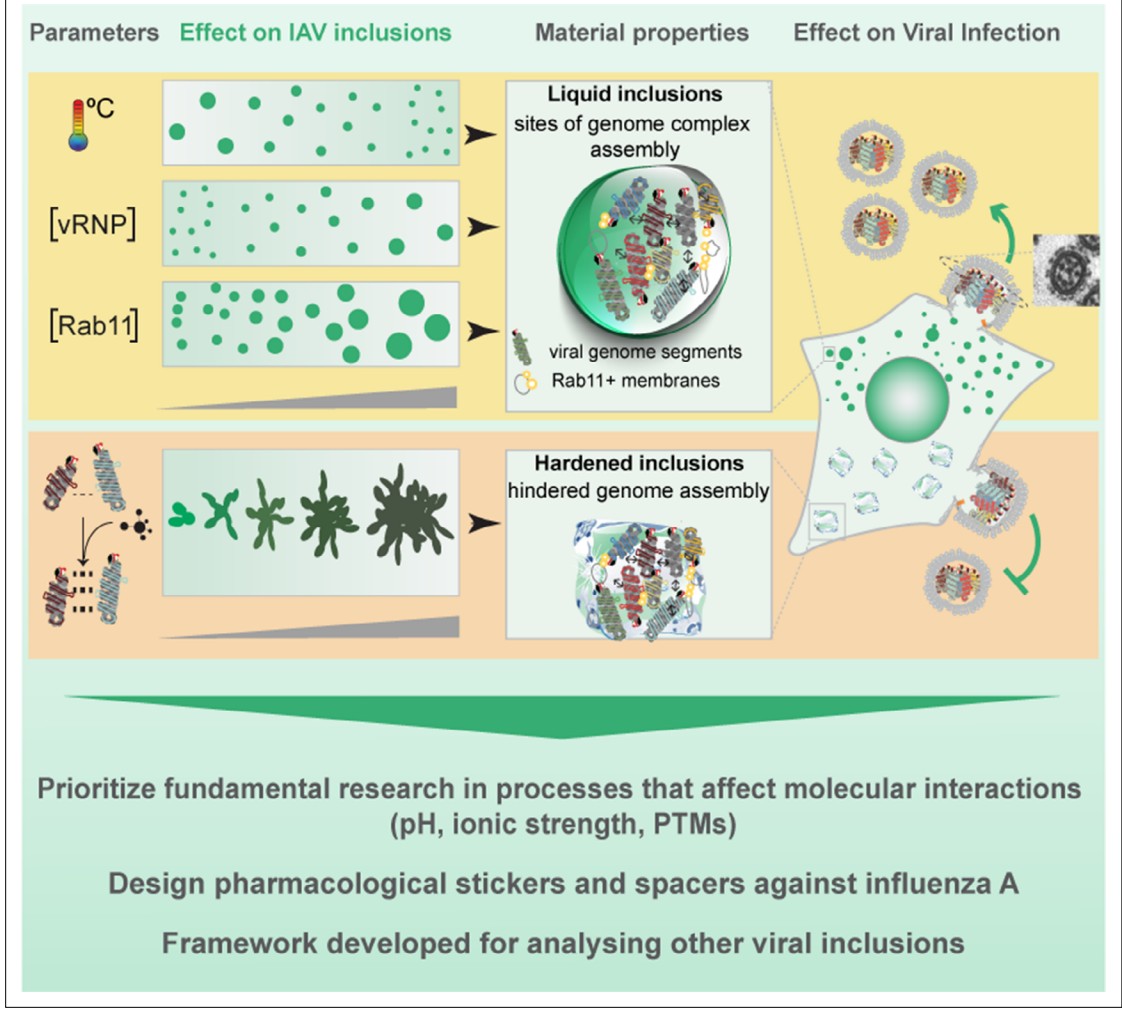

**Figure 10.** Working model for the strategies to modulate influenza A virus (IAV) liquid inclusions.

a caged incubator to control temperature at 37°C. An ROI inside the inclusion was pulsed with an excitation laser (455 nm) and internal rearrangement of the bleached region was monitored (chase) by live imaging for 120 s.

## Particle tracking and coarsening assay

Coarsening assays were calculated from two biological replicates in which several events were imaged. TrackMate plugin ((Fiji Is Just) ImageJ 2.1.0/1.53p, FIJI) was used to track inclusions at a timescale of 1 s/frame in live imaging samples and XY trajectories were subsequently analysed in a custom R (version 4.1.0) script. Using (FIJI and R), coarsening assay was analysed from time-lapsed tracking of two inclusions, starting from the point they first touch to the point they relax into a rounded puncta with an aspect ratio (AR) of 1.

## Solubility proteome profiling

Data was acquired from independent biological replicates of at least two experiments. A549, GFP-Rab11a-WT, and GFP-Rab11a-DN cells were mock-infected or infected with PR8 virus between 4 and 16 hpi and treated with nucleozin or DMSO. Frozen cell pellets containing $1\times10^6$ cells were shipped to Proteomics Core Facility at EMBL, Heidelberg, for further sample processing.

Samples for mass spectrometry analysis were prepared as described (*Zhang et al., 2022*). Briefly, $1\times10^6$ cells were resuspended in 100 µL lysis buffer (0.8%(v/v)) NP-40, 1× cOmplete protease inhibitor cocktail (Roche), 1× PhosphoStop (Roche), 1 U/mL RNAsin (Promega), 1.5 mM $MgCl_2$ in PBS

(2.67 mM KCl, 1.5 mM KH$_2$PO$_4$, 137 mM NaCl, and 8.1 mM NaH$_2$PO$_4$, pH 7.4). The sample aliquot for total proteome was incubated directly with benzonase on ice, while the sample aliquot for the soluble proteome was spun down at 100,000 × $g$ at 4°C for 20 min. The supernatant was incubated with benzonase. Both total and soluble aliquots were incubated for 10 min with final 1% SDS. Protein concentration was determined for the total proteome sample and aliquots equal to 5 µg protein were taken for sample preparation for MS analysis. Both soluble and total lysate of each sample was combined in a multiplexing MS experiment.

## Mass spectrometry sample preparation

Sample preparation for mass spectrometric measurements were performed as described in *Sridharan et al., 2019*; *Mateus et al., 2020*.

## Protein digestion and labelling

Protein digestion was performed using a modified SP3 protocol (*Hughes et al., 2014*; *Hughes et al., 2019*). Five µg of proteins (per condition) were diluted to a final volume of 20 µL with 0.5% (v/v) SDS and mixed with a bead slurry (Sera-Mag Speed beads, Thermo Fisher Scientific in ethanol) and incubated on a shaker at RT for 15 min. The beads were washed four times with 70% ethanol. Proteins on beads were overnight reduced (1.7 mM TECP), alkylated (5 mM chloroacetamide) and digested (0.2 µg trypsin, 0.2 µg LysC) 100 mM HEPES, pH 8. On the next day, peptides were eluted from the beads, dried under vacuum, reconstituted in 10 µL of water, and labelled with TMT-16plex reagents for 1 hr at RT. The labelling reaction was quenched with 4 µL of 5% (v/v) hydroxylamine and the conditions belonging to a single MS experiment were pooled together. The pooled sample was desalted with solid-phase extraction after acidification with 0.1% formic acid. The samples were loaded on a Waters OASIS HLB µelution plate (30 µm), washed twice with 0.05% formic acid, and finally eluted in 100 µL of 80% (v/v) acetonitrile containing 0.05% (v/v) formic acid. The desalted peptides were dried under vacuum and reconstituted in 20 mM ammonium formate. The samples were fractionated using C18-based reversed-phase chromatography running at high pH. Mobile phases constituted of 20 mM ammonium formate pH 10 (buffer A) and acetonitrile (buffer B). This system was run at 0.1 mL/min on the following gradient: 0% B for 0–2 min, linear increase 0–35% B in 2–60 min, 35–85% B in 60–62 min, maintain at 85% B until 68 min, linear decrease to 0% in 68–70 min, and finally equilibrated the system at 0% B until 85 min. Fractions were collected between 2 and 70 min and every 12th fraction was pooled together and vacuum dried.

## LC-MS-MS measurement

Samples were resuspended in 0.05% formic acid, 4% ACN in LC-MS grade water, and analysed on Q Exactive Plus mass spectrometer (Thermo Fisher Scientific) connected to UltiMate 3000 RSLC nano system (Thermo Fisher Scientific) equipped with a trapping cartridge (Precolumn; C18 PepMap 100, 5 µm, 300 µm ID×5 mm, 100 Å) and an analytical column (Waters nanoEase HSS C18 T3, 75 µm×25 cm, 1.8 µm, 100 Å) for chromatographic separation. Mobile phase constituted of 0.1% formic acid in LC-MS grade water (buffer A) and 0.1% formic acid in LC-MS grade acetonitrile (buffer B). The peptides were loaded on the trap column (30 µL/min of 0.05% (v/v) trifluoroacetic acid in LC-MS grade water for 3 min) and eluted using a gradient from 2% to 30% (v/v) buffer B over 103 min at 0.3 µL/min (followed by an increase to 40% B, and a final wash to 80% B for 2 min before re-equilibration to initial conditions). The outlet of the LC- system was directly fed for MS analysis using a Nanospray-Flex ion source and a Pico-Tip Emitter 360 µm OD×20 µm ID; 10 µm tip (New Objective). The mass spectrometer was operated in positive ion mode. The spray voltage and capillary temperature was set to 2.2 kV and 275°C, respectively. Full-scan MS spectra with a mass range of 375–1200 m/z were acquired in profile mode using a resolution of 70,000 (maximum fill time of 250 ms or a maximum of 3e6 ions (automatic gain control [AGC])). Fragmentation was triggered for the top 10 peaks with charge 2–4 on the MS scan (data-dependent acquisition) with a 30 s dynamic exclusion window (normalised collision energy was 30), and MS/MS spectra were acquired in profile mode with a resolution of 35,000 (maximum fill time of 120 ms or an AGC target of 2e5 ions).

## Protein identification and quantification

The MS data was processed as described in *Sridharan et al., 2019*. Briefly, the raw MS data was processed with isobarQuant and identification of peptides and proteins was performed with Mascot 2.4 (Matrix Science) against a database containing *Homo sapiens* Uniprot FASTA (proteome ID: UP000005640, downloaded on 14 May 2016) and Influenza A virus (strain A/Puerto Rico/8/1934 H1N1, proteome ID: UP000009255) along with known contaminants and the reverse protein sequences (search parameters: trypsin; missed cleavages 3; peptide tolerance 10 ppm; MS/MS tolerance 0.02 Da; fixed modifications included carbamidomethyl on cysteines and TMT16plex on lysine; variable modifications included acetylation of protein N-terminus, methionine oxidation, and TMT16plex on peptide N-termini).

## Mass spectrometry data analysis and normalisation

All MS data analysis was performed using R studio (version 1.2.1335 and R version 3.6.1). Data normalisation of NP40- and SDS-derived proteomes was performed with *vsn* (*Huber et al., 2002*). The overall signal sum intensities distributions from all TMT channels of all replicates were corrected for technical variations.

## Differential analysis of protein abundance

The $\log_2$ transformed *vsn* normalised SDS-derived signal sum intensities of proteins from different samples were analysed for differential abundances using *limma* (*Ritchie et al., 2015*). Proteins with $|\log_2(\text{fold change})| > 0.5$ and adjusted p-value (Benjamini Hochberg) <0.1 were considered significantly changed.

## Differential analysis of protein solubility

Solubility is defined as the ratio of NP40- and SDS-derived abundances of proteins. This ratio was computed for all proteins measured in a dataset. The $\log_2$ transformed protein solubility was compared between different conditions (timepoints of infection or different cell line at 12 hr post-infection) using *limma*. Proteins with $|\log_2(\text{fold change})| > 0.5$ and adjusted p-value (Benjamini Hochberg) <0.1 were considered significantly changed.

## Gene ontology overrepresentation analysis

Differential abundant or soluble human proteins from infection time course or different cell line datasets were used for GO term 'Biological processes' and/or 'Cellular compartments' overrepresentation analysis using clusterProfiler (R Bioconductor) (*Yu et al., 2012*). All identified proteins in each dataset served as the background. Standard settings were used for representing enriched GO terms (p-value cutoff: 0.05, Benjamini-Hochberg procedure for multiple testing adjustment and q-value cutoff of 0.2).

## Quantification and statistical analysis

Data were analysed using the R statistical package (R version 4.1.0), but visualisation of body weight changes was done in GraphPad Prism 9.4.1 (681). To quantify thermodynamics and topological variables, we extracted imaging data using an ImageJ custom plugin and a custom R analytics pipeline. For particle tracking, coarsening assay, photoactivation, photobleaching, and shock treatments, we compared two groups: cells treated with DMSO and nucleozin. In mice, comparison also involved the Mock- and PR8- infection. After data transformation in R, we assessed for homogeneity of variance. Homogenously distributed data were assessed by parametric test using either one-way ANOVA to analyse independent variables, followed by a post hoc analysis by Tukey multiple comparisons of means or t-test for comparison of two groups only. When the data is not homogenous, we used non-parametric analysis with statistical levels determined after Kruskal Wallis Bonferroni treatment. For simplicity, the details of the test used for each experiment are included in the figure legends. In our case, when two groups were compared, they were not homogenously distributed, hence a non-parametric analysis was done instead of a t-test. Alphabets above each boxplot represents the statistical differences between groups. Same alphabets indicate lack of significant difference between groups while different alphabets infer a statistically significant difference at α=0.05.

## Acknowledgements

This project has received funding from the European Research Council (ERC) under the European Union's Horizon 2020 research and innovation programme (grant agreement No. 101001521). Salary support from FCT: TAE, DB, VM are funded by PhD fellowships (PD/BD/128436/2017, PD/BD/148391/2019, and UI/BD/152254/2021) and SVC by DL 57. This work had support from SymbNET 'Genomics and Metabolomics in a Host-Microbe Symbiosis Network', funded by the European Union's Horizon 2020 research and innovation programme under grant agreement No 952537. The authors gratefully acknowledge the imaging, quantitative biology, and mouse facilities of the Instituto Gulbenkian de Ciência for their support and assistance in this work (especially Drs Gabriel Martins, Nuno Pimpão, Maria Hanulova, and Tiago Paixão).

## Additional information

### Funding

| Funder | Grant reference number | Author |
| --- | --- | --- |
| European Research Council | 101001521 | Maria-João Amorim |
| Fundação para a Ciência e a Tecnologia | PD/BD/128436/2017 | Temitope Akhigbe Etibor |
| European Commission Twinning Action Symbnet | 952537 | Maria-João Amorim |
| Fundação para a Ciência e a Tecnologia | PD/BD/148391/2019 | Daniela Brás |
| Fundação para a Ciência e a Tecnologia | UI/BD/152254/2021 | Victor Hugo Mello |

The funders had no role in study design, data collection and interpretation, or the decision to submit the work for publication.

### Author contributions

Temitope Akhigbe Etibor, Conceptualization, Data curation, Software, Formal analysis, Validation, Investigation, Visualization, Methodology, Writing - original draft, Writing - review and editing; Silvia Vale-Costa, Daniela Brás, Formal analysis, Validation, Investigation, Visualization, Methodology, Writing - review and editing; Sindhuja Sridharan, Conceptualization, Resources, Data curation, Software, Formal analysis, Validation, Investigation, Visualization, Methodology, Writing - review and editing; Isabelle Becher, Conceptualization, Data curation, Formal analysis, Validation, Visualization, Methodology, Writing - review and editing; Victor Hugo Mello, Resources, Data curation, Formal analysis, Validation, Visualization; Filipe Ferreira, Resources, Data curation, Formal analysis, Validation, Methodology, Writing - review and editing; Marta Alenquer, Formal analysis, Validation, Writing - review and editing; Mikhail M Savitski, Conceptualization, Resources, Data curation, Supervision, Funding acquisition, Project administration, Writing - review and editing; Maria-João Amorim, Conceptualization, Resources, Data curation, Formal analysis, Supervision, Funding acquisition, Validation, Investigation, Visualization, Methodology, Writing - original draft, Project administration, Writing - review and editing

### Author ORCIDs

Temitope Akhigbe Etibor  http://orcid.org/0000-0002-9024-4310
Isabelle Becher  http://orcid.org/0000-0001-7170-2235
Marta Alenquer  http://orcid.org/0000-0003-3062-1222
Mikhail M Savitski  http://orcid.org/0000-0003-2011-9247
Maria-João Amorim  http://orcid.org/0000-0002-4129-6659

### Ethics

Animals were group housed in individually ventilated cages with access to food and water ad libitum. This research project was ethically reviewed and approved by both the Ethics Committee and the

Animal Welfare Body of the IGC (license reference: A003.2021), and by the Portuguese National Entity that regulates the use of laboratory animals DGAV - Direção Geral de Alimentação e Veterinária (license references: 0421/000/000/2022, Controlling influenza A virus liquid organelles - LOFLU, funded by the European Research Council). All experiments conducted on animals followed the Portuguese (Decreto-Lei n° 113/2013) and European (Directive 2010/63/EU) legislations, concerning housing, husbandry, and animal welfare.

### Decision letter and Author response
Decision letter https://doi.org/10.7554/eLife.85182.sa1
Author response https://doi.org/10.7554/eLife.85182.sa2

## Additional files

### Supplementary files
• Supplementary file 1. Table with values for all thermodynamic related parameters. (Sheet 1): Topology, thermodynamics, and material properties of influenza A virus (IAV) inclusions. After PR8 infection (1 MOI), cells were subjected to thermal changes. (Sheets 2–3): Topology, thermodynamics, and material properties of IAV inclusions. After PR8 infection (1 MOI) inclusions were analysed at different hours post-infection (hpi) as viral ribonucleoprotein (vRNP) concentration increases (Sheet 2) or with overexpressed Rab11 (Sheet 3). (Sheet 4): Topology, thermodynamics, and material properties of IAV inclusions. After PR8 infection (1 MOI) inclusions were subjected to nucleozin or dimethyl sulfoxide (DMSO) treatment in WT or the mutant virus NP-Y289H. (Sheet 5): Topology, thermodynamics, and material properties of IAV inclusions. After PR8 infection (1 MOI) inclusions were analysed upon FLAPh in native and overexpressing conditions of Rab11. (Sheets 6–8, 11)1: Topology, thermodynamics, and material properties of IAV inclusions. After PR8 infection (1 MOI) inclusions were subjected to live imaging for inclusion tracking (Sheet 6), fusion dynamics (Sheet 7), and FLAPh (Sheet 8). Assessment of the biophysical traits of inclusions upon DMSO or nucleozin treatment subjected to hexanediol and hypotonic shock treatments is shown in (Sheet 11). (Sheets 9–10): Topology, thermodynamics, and material properties of IAV inclusions. Mice was infected with X31 and treated with PBS or nucleozin for the analysis of number of inclusion and topology in lung slices (Sheet 9) and body weight (Sheet 10).

• Supplementary file 2. Differential analysis of protein solubility changes before and after nucleozin treatment at 12 hr post-infection in WT and Rab11a-DN cell lines. PR8-infected Rab11a-WT and Rab11a-DN cells were treated with either dimethyl sulfoxide (DMSO) (vehicle) or 5 µM of nucleozin for 1 hr. The protein solubility changes upon nucleozin (or DMSO) treatment in RAB11a-WT and Rab11a-DN cells is listed in this table.

• Supplementary file 3. Differential analysis of protein abundance changes before and after nucleozin treatment at 12 hr post-infection in WT and Rab11a-DN cell lines.

• MDAR checklist

### Data availability
The mass spectrometry proteomics data have been deposited to the ProteomeXchange Consortium via the PRIDE (PubMed ID: 34723319) partner repository with the dataset identifier PXD041748. Computer code or algorithm used to generate the results reported in the paper are available at https://doi.org/10.5281/zenodo.7709159. All data for Figures and Figure supplements are provided within this paper and at https://doi.org/10.5281/zenodo.7709159. Sequences of described viruses are accessible from the NCBI virus under accession number GCF_000865725.1.

The following datasets were generated:

| Author(s) | Year | Dataset title | Dataset URL | Database and Identifier |
|---|---|---|---|---|
| Amorim MJ | 2023 | Proteome wide solubility profiling of Nucleozin treated A549 cells infected with Influenza-A | https://www.ebi.ac.uk/pride/archive/projects/PXD041748 | PRIDE, PXD041748 |
| Etibor TA, Vale-Costa S, Sridharan S, Brás D, Becher I, Mello VH, Ferreira F, Alenquer M, Savitski MM, Amorim MJ | 2023 | Defining basic rules for hardening influenza A virus liquid condensates | https://doi.org/10.5281/zenodo.7709159 | Zenodo, 10.5281/zenodo.7709159 |

The following previously published dataset was used:

| Author(s) | Year | Dataset title | Dataset URL | Database and Identifier |
|---|---|---|---|---|
| ViralMulti | 2000 | ViralMultiSegProj15521 | https://www.ncbi.nlm.nih.gov/assembly/GCF_000865725.1/ | NCBI Assembly, GCF_000865725.1 |

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
