## [Editor Report]

Etibor and collaborators have performed a series of well-thought and careful experiments to understand some of the physical and thermodynamic properties of liquid condensates produced by the infection with the influenza A virus. However, their approach and rules could be easily applied to any other cellular phenomena that involve the formation of intracellular liquid condensates. Finally, this article is setting up the basis for an in-depth theoretical analysis of the physical phenomena described here and their correlation with the biology of intracellular liquid condensates.

---

## [Decision Letter]

**Decision letter after peer review:**

Thank you for submitting your article "Rules for hardening influenza A virus liquid condensates" for consideration by *eLife*. Your article has been reviewed by three peer reviewers, including Mauricio Comas-Garcia as the Reviewing Editor and Reviewer #1, and the evaluation has been overseen by Anna Akhmanova as the Senior Editor.

Essential revisions:

1) The authors should check and review extensively for improvements to the use of English.

2) Please address all the comments from reviewers 1 and 3, and the minor comments from reviewer 2.

3) Please address at least the major comments 1, 2, and 4 from reviewer 2. It would be ideal for addressing major comment 3, but if these experiments cannot be performed on time, please add a rational explanation to solve this particular question,

*Reviewer #1 (Recommendations for the authors):*

I enjoyed your article. The ideas are extremely interesting. However, it is extremely hard to read. Although I mentioned this before, this problem does not arise from technical flaws but is most likely from a language barrier. Having this problem solved and defining most of the terms you discuss in the results in the introduction would make the manuscript readable.

I would strongly suggest collaborating with a theoretical physicist or a physical chemist, as I am sure that the data you have presented here is the tip of the iceberg for a more in-deep analysis of the thermodynamics of the viral liquid condensates. It is my opinion that now the data looks very cool and can provide some insight into what is going on, but there is a lack of a much deeper analysis of the thermodynamics processes that you are describing.

I know this part can be repetitive with respect to what I wrote in the public version, but I see this manuscript as a diamond in the rough, and with some further analysis, it could become a gem.

*Reviewer #2 (Recommendations for the authors):*

The following suggestions are here to improve the manuscript.

1) It is not clear to the reviewer how ΔΔG is calculated. In line 879, the authors wrote, "The change in free energy was normalised to 3hpi (an infection stage with nuclear vRNP staining lacking cytoplasmic inclusions) and was represented as ΔΔG = ΔG – ΔG(3 hpi)". However, how can they calculate ΔG(3 hpi) as G=-RTLnK with K=Cdense/Cdilute and Cdense=0 after 3h?

2) Similarly, in table supplement 1, in sheet 1 to 4, there is "C_dil" and "total_C_dil" which have different values. However, total Cdil is indicated as Cdilute in the description of the parameters. What is the difference between both parameters and which one is used to calculate K?

3) I would suggest performing more systematic FRAP/FLAPh experiments on cells expressing fluorescent versions of both NP and Rab11a to investigate the influence of condensate size, time after infection, or global concentrations of Rab11a in the cell (using the total fluorescence of overexpressed GFP-Rab11a as a proxy) on condensate properties.

4) Eventual artifacts induced by the use of antibodies to quantify Cdense should be mentioned in the paragraph dedicated to the limitations of the study.

*Reviewer #3 (Recommendations for the authors):*

The study was well-carried out, and the implications could be huge – congrats on the great work! A question that I was wondering – have you tried to obtain a drug-resistant mutant of nucleozin, and see where the mutation(s) are? (c.f. Risso-Ballester et al., 2021) That could give important insights into how nucleozin works at the molecular level.

---

## [Author Response]

Reviewer #1 (Recommendations for the authors):I enjoyed your article. The ideas are extremely interesting. However, it is extremely hard to read. Although I mentioned this before, this problem does not arise from technical flaws but is most likely from a language barrier. Having this problem solved and defining most of the terms you discuss in the results in the introduction would make the manuscript readable.I would strongly suggest collaborating with a theoretical physicist or a physical chemist, as I am sure that the data you have presented here is the tip of the iceberg for a more in-deep analysis of the thermodynamics of the viral liquid condensates. It is my opinion that now the data looks very cool and can provide some insight into what is going on, but there is a lack of a much deeper analysis of the thermodynamics processes that you are describing.I know this part can be repetitive with respect to what I wrote in the public version, but I see this manuscript as a diamond in the rough, and with some further analysis, it could become a gem.

We apologise for not being able at this point to provide more thermodynamic parameters (such as entropy and enthalpy) or physic models despite having teamed up with the theoretical physicist Pablo Sartori. For the future we aim to combine different vRNPs in cells and in the developed in vitro reconstitution system to derive meaningful principles operating in the assembly of influenza A virus genome.

Reviewer #2 (Recommendations for the authors):The following suggestions are here to improve the manuscript.1) It is not clear to the reviewer how ΔΔG is calculated. In line 879, the authors wrote, "The change in free energy was normalised to 3hpi (an infection stage with nuclear vRNP staining lacking cytoplasmic inclusions) and was represented as ΔΔG = ΔG – ΔG(3 hpi)". However, how can they calculate ΔG(3 hpi) as G=-RTLnK with K=Cdense/Cdilute and Cdense=0 after 3h?

To calculate the ΔG, we used method published in Riback *et al.*, 2020. Here, ΔG = RTLnK, where is K is the partition coefficient calculated as C_dense_ / C_dilute_. After calculating the ΔG at all time points, each ΔG was normalised to the time when inclusions have not yet emerged in the cytosol (but vRNPs begin to reach the cytosol), which is 3 hpi. At 3hpi, vRNPs are homogenously distributed in the cytosol without visible inclusions, serving as good control for when there is no phase transition. Here, as inclusions were not formed, we designed an ImageJ macro to collect ROI from Rab11 channel and superimpose it on NP channel (with NP as a proxy for vRNPs). The mean intensities within and outside these NP ROIs were analysed as C_dense_ and C_dilute_ respectively for 3 hpi. Ideally, in the absence of phase transition, C_dense_ = C_dilute_, K = 1, and ΔG = 0. However, at 3 hpi, K was either a few fractions above or below 1, with a median of 1.748089 and 1.689856 for GFP and GFP-Rab11a expressing cells respectively and median ΔG of -1439.5 and -1352.18 for GFP and GFP-Rab11a expressing cells respectively. This may be due to the computer generated C_dense_ ROI, which were contingent on Rab11 distribution (akin to mock infection) but different from Rab11-positive inclusions. Due to the variation in K and the magnitude of its effect in ΔG, we transformed ΔG to ΔΔG by normalising to ΔG at 3hpi using the formula: ΔΔG = ΔG – ΔG _(3 hpi)_. The exception is at different temperatures (where we showed the ΔG) which were all done at 8hpi and does not require normalization.

2) Similarly, in table supplement 1, in sheet 1 to 4, there is "C_dil" and "total_C_dil" which have different values. However, total Cdil is indicated as Cdilute in the description of the parameters. What is the difference between both parameters and which one is used to calculate K?

The values used are the ones in the described parameters. In Materials and methods, we explained the possible methods for calculating K and in our case two methods were readily feasible and made more sense.

To calculate K, C_dense_ and C_dilute_ must first be extrapolated. C_dense_ was calculated as the mean intensity of each inclusion per cell (i.e. Nth C_dense_ per cell). To calculate Cdilute three strategies are possible: (1) As published by Riback *et al.* 2020, a number of C_dilute_ ROIs were selected outside the C_dense_. This method is not feasible in our situation as inclusions were ubiquitous in the cytoplasm and presents a technical challenge. (2) Using a custom macro, we measured C_dilute_ from ROI bands around every inclusion in the cytoplasm (denoted as C_dil in the supplementary table). This would have been the ideal method in our case as every C_dense_ would have its own corresponding C_dilute_. However, due to the ubiquity of inclusions in each cell, there many ROI bands overlapping with other inclusion region or another C_dilute_ region (as exemplified in Author response image 1). Even when we removed the inclusions before measuring the intensity of the ROI band, same problem persists in the opposite direction due to overlap with black filled spots (as exemplified in the figure 3 below). (3) Alternatively, we designed an analytic pipeline after measuring the C_dense_, we removed the inclusions and measured the remaining bulk cytoplasmic intensity as the C_dilute_ (denoted as total_C_dil in the supplementary data).

Finally, to pick the better of methods (2) and (3), we fitted their partition coefficient (K) to K calculated from median of all C_dense_/ total_C_dil. The correlated method (3) was then used for all analysis in this manuscript. In our case, the best method required the use of total_C_dil (a global but singular C_dilute_ i.e 1 C_dilute_ per cell) instead of C_dil (from around each viral inclusion per cell.i.e. N^th^ C_dilute_ per cell)

**Author response image 1. sa2fig1:** Sample segmentation of viral inclusions and milieu to measure the C_dense_ and C_dilute_ respectively.

3) I would suggest performing more systematic FRAP/FLAPh experiments on cells expressing fluorescent versions of both NP and Rab11a to investigate the influence of condensate size, time after infection, or global concentrations of Rab11a in the cell (using the total fluorescence of overexpressed GFP-Rab11a as a proxy) on condensate properties.

We have included a new figure, figure 5 with the suggested data.

4) Eventual artifacts induced by the use of antibodies to quantify Cdense should be mentioned in the paragraph dedicated to the limitations of the study.

We have discussed the problems of using antibodies or the PA-mNeonGreen PR8 virus in the limitations of the study.

Reviewer #3 (Recommendations for the authors):The study was well-carried out, and the implications could be huge – congrats on the great work! A question that I was wondering – have you tried to obtain a drug-resistant mutant of nucleozin, and see where the mutation(s) are? (c.f. Risso-Ballester et al., 2021) That could give important insights into how nucleozin works at the molecular level.

Using fluorescence spectroscopy, docking model, molecular cloning, and immunofluorescence microscopy, Kao *et al.* 2010, postulated that viral protein NP is critical to vRNP-nucleozin binding. Importantly, tyrosine residue Y289 of wild-type NP was shown to form hydrophobic interactions with nucleozin by aromatic stacking, resulting in vRNP aggregation. NP point mutation replacing tyrosine with histidine in the 289 position leads to the formation of an escape mutant (Y289H) unable to induce vRNP aggregation. Of note, Amorim *et al.*, 2013 showed by immunofluorescence microscopy, live imaging, and affinity purification (coupled with primer extension assay and western blot) that cytoplasmic vRNP aggregation was lost cells infected with the Y289H nucleozin resistant strain.

In the present manuscript, we have now confirmed the data that the NP mutant Y289H is resistant to nucleozin and applied our thermodynamic analyses to cells infected with it. The results are shown in Figure 4 P- R of the manuscript.